https://doi.org/10.1038/s41467-019-10329-3　　**OPEN**

# Selective recruitment of different $Ca^{2+}$-dependent transcription factors by STIM1-Orai1 channel clusters

Yu-Ping Lin[1], Daniel Bakowski[1], Gary R. Mirams ⓘ [2] & Anant B. Parekh[1]

Store-operated $Ca^{2+}$ entry, involving endoplasmic reticulum $Ca^{2+}$ sensing STIM proteins and plasma membrane Orai1 channels, is a widespread and evolutionary conserved $Ca^{2+}$ influx pathway. This form of $Ca^{2+}$ influx occurs at discrete loci where peripheral endoplasmic reticulum juxtaposes the plasma membrane. Stimulation evokes numerous STIM1-Orai1 clusters but whether distinct signal transduction pathways require different cluster numbers is unknown. Here, we show that two $Ca^{2+}$-dependent transcription factors, NFAT1 and c-fos, have different requirements for the number of STIM1-Orai1 clusters and on the $Ca^{2+}$ flux through them. NFAT activation requires fewer clusters and is more robustly activated than c-fos by low concentrations of agonist. For similar cluster numbers, transcription factor recruitment occurs sequentially, arising from intrinsic differences in $Ca^{2+}$ sensitivities. Variations in the number of STIM1-Orai1 clusters and $Ca^{2+}$ flux through them regulate the robustness of signalling to the nucleus whilst imparting a mechanism for selective recruitment of different $Ca^{2+}$-dependent transcription factors.

---

[1] Department of Physiology, Anatomy and Genetics, Oxford University, Parks Road, Oxford OX1 3PT, UK. [2] Centre for Mathematical Medicine and Biology, School of Mathematical Sciences, Nottingham University, Nottingham NG7 2RD, UK. Correspondence and requests for materials should be addressed to A.B.P. (email: anant.parekh@dpag.ox.ac.uk)

Formation of signalling protein hubs physically associated with membranes is an evolutionarily conserved mechanism throughout the taxonomic ranks that serves to increase the rate and efficiency of transduction pathways. In geometrically complex or polarized cells such as neurons and pancreatic acini, signalling complexes maintain asymmetric vectorial ion transport and compartmentalised responses[1]. Architecturally simpler cells are also punctuated with membrane signalling complexes, suggesting these clusters have a fundamental role in signal transduction.

Store-operated $Ca^{2+}$ release-activated $Ca^{2+}$ (CRAC) channels represent a striking example of membrane protein clustering. CRAC channels activate upon a loss of $Ca^{2+}$ from within the lumen of the endoplasmic reticulum (ER), which occurs following stimulation of cell-surface receptors that generate the $Ca^{2+}$-releasing second messenger inositol trisphosphate[2]. The fall in ER $Ca^{2+}$ is directly sensed by STIM proteins[3,4], which then oligomerise[5,6] and translocate to within 20 nm of the plasma membrane. At these ER-plasma membrane junctions, clusters of STIM protein tether to and gate the plasma membrane protein Orai1, the pore-forming subunit of the CRAC channel[7–9]. The ensuing $Ca^{2+}$ flux controls a salmagundi of responses, including exocytosis, metabolism, motility, gene expression and cell growth and differentiation[10].

$Ca^{2+}$ microdomains near open CRAC channels activate downstream signalling pathways, including certain isoforms of adenylyl cyclase[11], plasma membrane $Ca^{2+}$ ATPase pumps[12], $Ca^{2+}$-dependent phospholipase $A_2$[13] and endothelial NO synthase[14]. These enzymes are located close to the CRAC channel and, at least for adenylyl cyclase isoform type 8, direct binding to Orai1 has been reported[11]. $Ca^{2+}$ microdomains near CRAC channels also increase gene expression through activation of $Ca^{2+}$-dependent transcription factors c-fos and nuclear factor of activated T cells (NFAT). Transcription of the immediate early gene c-fos is stimulated following activation of the non-receptor tyrosine kinase Syk by local $Ca^{2+}$ entry through CRAC channels[15]. Co-immunoprecipitation and immunocytochemical studies demonstrate that Syk associates with Orai1[15,16]. Syk phosphorylates the transcription factor STAT5[15], which dimerises and migrates to the nucleus to regulate c-fos transcription[17]. NFAT is extensively phosphorylated at rest and trapped within the cytosol[18]. Dephosphorylation by the $Ca^{2+}$-activated phosphatase calcineurin exposes a nuclear localisation sequence, enabling NFAT migration into the nucleus[18]. A fraction of cellular NFAT and calcineurin are held at the plasma membrane through association with the scaffolding protein AKAP-79[19,20]. Upon store depletion, AKAP-79 interacts with Orai1, resulting in activation of calcineurin by $Ca^{2+}$ microdomains near the open channels.

The autonomous assembly of functional CRAC channels from STIM1 and Orai1 components into numerous yet discrete puncta raises two important questions. First, is there a signalling advantage conferred by channel clustering over a similar number of dispersed channels? Second, for the same rise in bulk cytosolic $Ca^{2+}$, are fewer puncta each with large $Ca^{2+}$ influx equally effective in activating signalling pathways as many puncta with reduced $Ca^{2+}$ flux? Previous work has shown that, for a similar number of functional CRAC channels, channel clustering leads to more robust activation of NFAT and c-fos transcription factors[16]. Here, we address the second question. We find, for the same bulk $Ca^{2+}$ rise, reduced $Ca^{2+}$ flux through a large number of STIM-Orai1 puncta is considerably more effective in activating NFAT1 than larger flux through fewer puncta. Additionally, for similar number of puncta, larger $Ca^{2+}$ flux is required to stimulate c-fos. Our data reveal that different signalling pathways can be recruited by variations in the number of channel clusters formed and, for

similar numbers of clusters, by the extent of $Ca^{2+}$ flux through the channels.

## Results

**Matching $Ca^{2+}$ entry to different levels of store depletion.** Once store $Ca^{2+}$ content falls below a threshold[21], STIM1 redistribution and CRAC current increase as a function of store depletion (depicted schematically in Fig. 1a)[22,23]. $Ca^{2+}$ flux is determined principally by the electrochemical gradient. At negative potentials, the electrical gradient is large but falls supralinearly as membrane potential depolarises (Fig. 1b). We therefore reasoned that modest store depletion under hyperpolarised conditions in RBL cells should produce an equivalent rise in bulk cytosolic $Ca^{2+}$ as strong store depletion under depolarised conditions, because large $Ca^{2+}$ flux through few channels should raise bulk $Ca^{2+}$ to a similar extent as reduced flux through many channels. A simple tool to deplete stores to varying extents is thapsigargin, an inhibitor of the Sarco-Endoplasmic Reticulum $Ca^{2+}$ATPase (SERCA) pump[24]. Strong store depletion with a high dose of thapsigargin leads to formation of numerous STIM1 puncta below the plasma membrane whereas weaker store depletion evokes less accumulation of peripheral STIM1[22]. The electrical driving force can be manipulated by varying the external $K^+$ concentration, since RBL cells express numerous inwardly rectifying $K^+$ channels[25]. In standard $K^+$ (2.8 mM) solution, the membrane potential is ~ −80 mV but depolarises close to ~0 mV in high $K^+$ (100 mM) solution. Therefore, a high dose of thapsigargin in high $K^+$ solution will induce numerous STIM1-Orai1 puncta but unitary $Ca^{2+}$ flux will be low. Conversely, in the presence of a low thapsigargin concentration in standard external $K^+$ solution, fewer puncta will form but the unitary $Ca^{2+}$ flux will be relatively large. Careful selection of appropriate thapsigargin concentrations should, therefore, enable comparison of the signalling power of few versus many puncta, with both producing the same rise in bulk cytosolic $Ca^{2+}$ (Fig. 1c). We therefore designed experiments to identify a suitable dose of thapsigargin.

We applied different concentrations of thapsigargin in $Ca^{2+}$-free solution to measure dose-dependent $Ca^{2+}$ release from the stores (Fig. 1d). Readmission of external $Ca^{2+}$ resulted in store-operated $Ca^{2+}$ influx (Fig. 1d). The relationship between the rate of $Ca^{2+}$ influx and thapsigargin concentration is shown in Fig. 1e. A fit of the curve revealed a Hill coefficient of 0.8 and an apparent $K_D$ of 0.025 μM. Application of 100 nM thapsigargin in high $K^+$ solution evoked similar $Ca^{2+}$ release from the stores as that elicited by 100 nM thapsigargin in standard $K^+$ solution (Fig. 1d), as expected for similar extents of store depletion. However, in high $K^+$ solution, the rate of $Ca^{2+}$ influx slowed considerably (Fig. 1d) and was similar to that evoked by 30 nM thapsigargin in standard $K^+$ solution (Fig. 1e). To confirm that 100 nM thapsigargin in high $K^+$ solution led to more store depletion than 30 nM thapsigargin, we analysed the ionomycin-evoked cytosolic $Ca^{2+}$ transient as a proxy for store $Ca^{2+}$ content[23]. Whereas application of ionomycin led to a large $Ca^{2+}$ transient in cells exposed to $Ca^{2+}$-free solution alone (Fig. 1f), the response was progressively reduced by prior challenge with different concentrations of thapsigargin (Fig. 1f). We measured the initial rate of rise of cytosolic $Ca^{2+}$ following ionomycin challenge in $Ca^{2+}$-free solution as this better reflects the free $Ca^{2+}$ levels within the store. Two micromolar thapsigargin almost fully suppressed the subsequent ionomycin response, demonstrating that the thapsigargin- and ionomycin-sensitive stores overlap substantially. Stimulation with 100 nM thapsigargin reduced the ionomycin response by >80% as did stimulation with 100 nM thapsigargin in high $K^+$ solution. However, pre-treatment with

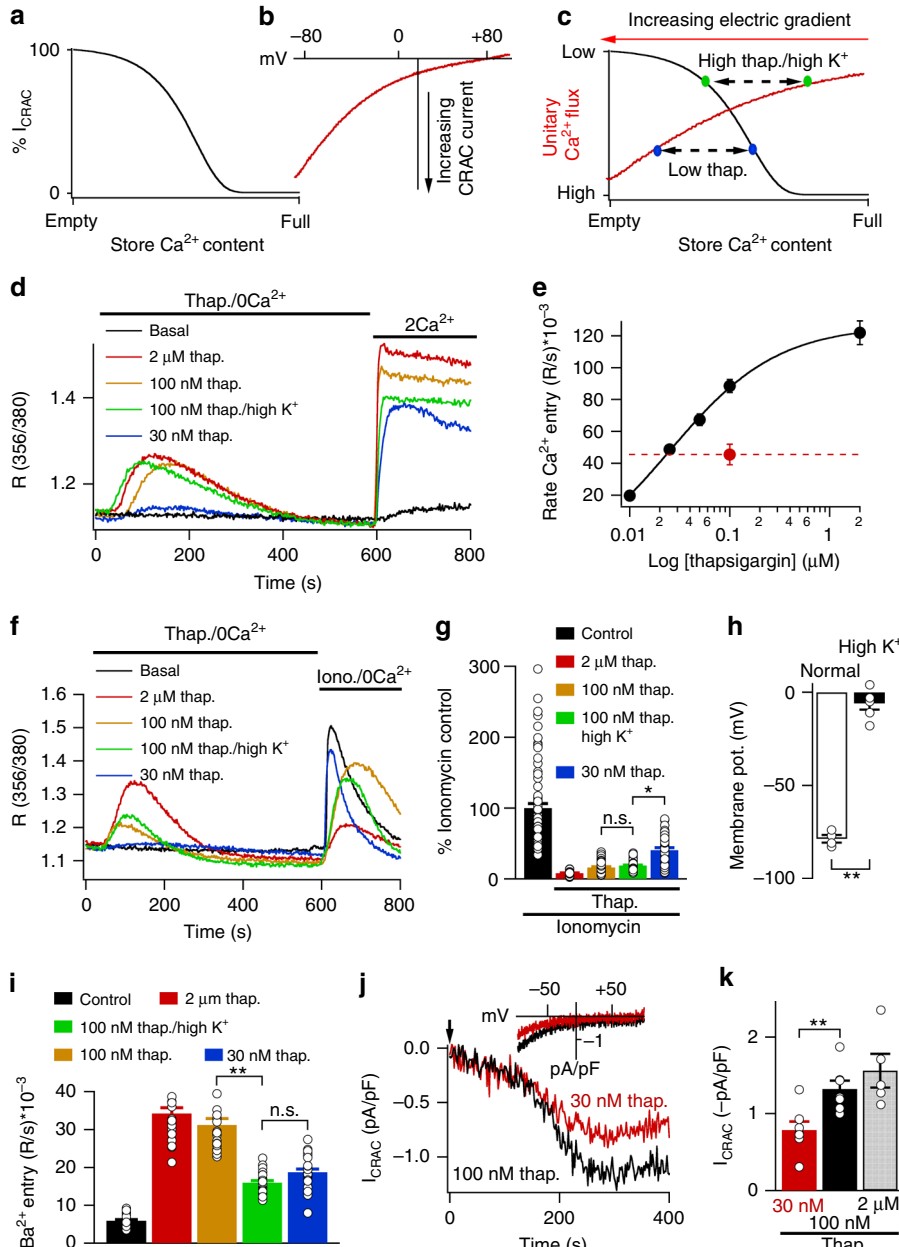

**Fig. 1** Matching cytosolic $Ca^{2+}$ rises to low and high thapsigargin concentrations. **a** Schematic depicts the relationship between store $Ca^{2+}$ content and $I_{CRAC}$ activation, based on ref. [22]. It is included for demonstrative purposes only. **b** Current–voltage relationship for CRAC current from an RBL cell. Pipette solution contained 30 μM $InsP_3$ and 10 mM EGTA to deplete the stores. **c** The graph combines panels **a** and **b**, to show the relationships between store depletion, electrical gradient and unitary $Ca^{2+}$ flux. **d** $Ca^{2+}$ release and $Ca^{2+}$ influx are shown following stimulation with different concentrations of thapsigargin. Basal depicts cells exposed to the same solutions but without thapsigargin. **e** Graph compares the rate of rise of cytosolic $Ca^{2+}$ after readmission of external $Ca^{2+}$ following stimulation with different concentrations of thapsigargin. Each point is the mean of 11–17 cells. Basal rate was subtracted. Red dot indicates $Ca^{2+}$ entry rate to 100 nM thapsigargin in high $K^+$ solution. **f** Ionomycin (2 μM)-evoked $Ca^{2+}$ release is compared between cells treated with different concentrations of thapsigargin. **g** Aggregate data from experiments as in panel **f** are compared. Control denotes cells that were not challenged with thapsigargin prior to ionomycin stimulation. Each bar is the mean of 44–82 cells. **h** Membrane potential is compared in cells exposed to standard (2.8 mM) $K^+$ or high (100 mM) $K^+$ solution, measured in current clamp mode. Each bar denotes data from eight cells. **i** $Ba^{2+}$ entry rate is compared for the conditions shown. Cells were stimulated with different concentrations of thapsigargin in $Ca^{2+}$-free solution and then 2 mM $Ba^{2+}$ was applied. Each bar is the mean of between 22 and 31 cells. **j** Whole-cell patch clamp experiments compare $I_{CRAC}$ following activation with 30 nM or 100 nM thapsigargin. Thapsigargin was applied at the arrow. Inset compares I–V relationships (taken when the currents had peaked) for the two stimuli. **k** Bar chart compares mean data for the conditions shown. Thirty nanomolar and 100 nM bars are the mean of seven cells each and 2 μM is the mean of five cells. *denotes $p < 0.05$, **$p < 0.01$ and n.s. not significant, determined using unpaired Student's $t$-test. Error bars denote SEM

30 nM thapsigargin led to ~2-fold less store depletion (Fig. 1g). One-hundred nanomolar thapsigargin in high $K^+$ solution, therefore, evokes greater store depletion than 30 nM thapsigargin in standard $K^+$ solution.

We considered the possibility that, in the continuous presence of external $Ca^{2+}$, some store refilling indeed occurred in cells challenged with 30 nM thapsigargin as this is a sub-maximal concentration and therefore not all SERCA pumps would be

blocked. To test this, we compared store $Ca^{2+}$ content between cells challenged with 30 nM thapsigargin in the presence of external $Ca^{2+}$ for 40 min (prior to stimulation with ionomycin in $Ca^{2+}$-free solution) with cells exposed to 30 nM thapsigargin for the same time but in the continuous presence of $Ca^{2+}$-free solution (Supplementary Fig. 1). Store $Ca^{2+}$ content was similar in both cases, confirming little refilling in cells challenged with 30 nM thapsigargin in the continuous presence of external $Ca^{2+}$.

In high $K^+$ solution, the electrical driving force for $Ca^{2+}$ entry will be reduced. To confirm this, we carried out current clamp recordings of the resting membrane potential in RBL cells. In standard $K^+$ solution, the resting membrane potential was ~ −80 mV (Fig. 1h) but this was depolarised close to 0 mV in high $K^+$ solution (Fig. 1h). $Ba^{2+}$ permeates CRAC channels and binds to fura-2 but is not transported out of the cell by $Ca^{2+}$-ATPase pumps, thereby providing a reasonable estimate of the rate of store-operated divalent cation influx, and therefore CRAC channel activity, in intact cells[26]. Following stimulation with different concentrations of thapsigargin in $Ca^{2+}$-free solution, we applied $Ba^{2+}$ externally and measured the rate of rise of the $Ba^{2+}$-induced fluorescence signal (Fig. 1i). The rise in $Ba^{2+}$ signal evoked by 100 nM thapsigargin in high $K^+$ solution was very similar to that induced by 30 nM thapsigargin in standard $K^+$ solution, consistent with both stimuli eliciting similar rates of divalent cation entry.

We also carried out whole-cell patch clamp experiments to compare directly $Ca^{2+}$ flux through CRAC channels following stimulation with either 30 nM or 100 nM thapsigargin. Challenge with 30 nM thapsigargin activated $I_{CRAC}$ after a delay (Fig. 1j; inset shows I–V relationship taken once the current had reached a peak). Stimulation with 100 nM thapsigargin led to the activation of a larger current (Fig. 1j). $I_{CRAC}$ induced by 30 nM thapsigargin was significantly smaller than that seen in response to 100 nM thapsigargin (Fig. 1k).

Finally, we compared the $Ca^{2+}$ signals evoked by different thapsigargin concentrations in the continuous presence of external $Ca^{2+}$ for 40 min, the time when we measured transcription factor activation (see below). Two micromolar or 100 nM thapsigargin elicited relatively large and prolonged elevations in cytosolic $Ca^{2+}$ whereas $Ca^{2+}$ signals to 30 nM thapsigargin in standard $K^+$ solution or 100 nM thapsigargin in high $K^+$ external solution were significantly smaller (Supplementary Fig. 2). The cytosolic $Ca^{2+}$ signals to 30 nM thapsigargin in standard $K^+$ or 100 nM thapsigargin in high $K^+$ solution were almost identical for up to 40 min of recording (Supplementary Fig. 2).

Therefore, the combination of 100 nM thapsigargin in high $K^+$ solution evokes flux through CRAC channels that raises bulk cytosolic $Ca^{2+}$ to a similar extent to that seen following stimulation with 30 nM thapsigargin in standard $K^+$ solution.

**STIM1 puncta increase with thapsigargin concentration.** We tested whether 100 nM thapsigargin in high $K^+$ solution evoked more STIM1 puncta than 30 nM thapsigargin in standard $K^+$ solution. After stimulation of RBL cells with different concentrations of thapsigargin for 10 min, cells were fixed and endogenous STIM1 stained with a monoclonal antibody. Numerous puncta were evoked following stimulation with either 2 μM thapsigargin in standard $K^+$ solution or 100 nM thapsigargin in high $K^+$ solution (Fig. 2a, b; no significant difference was found between the two groups). By contrast, stimulation with 30 nM thapsigargin led to far fewer STIM1 puncta (Fig. 2a, b).

We also expressed STIM1-YFP and measured puncta number using live cell imaging with confocal microscopy (Fig. 2c). As was

the case with endogenous STIM1 (Fig. 2a, b), the number of puncta increased with thapsigargin concentration (Fig. 2c, d).

We monitored the stability of puncta over 40 min, as this was the time over which we measured transcription factor activation (see below). Following stimulation with thapsigargin in the presence of external $Ca^{2+}$, STIM1 puncta were relatively stable over this time frame regardless of stimulus intensity (Fig. 2e).

**Dependence of NFAT1 migration on thapsigargin concentration.** Confocal images showed that NFAT1-GFP was located mainly in the cytosol at rest but stimulation with a supramaximal concentration of thapsigargin (2 μM) led to robust nuclear accumulation within 40 min (Fig. 3a; aggregate data are shown in Fig. 3b). Application of 100 nM thapsigargin also led to robust nuclear migration (Fig. 3a, b). However, exposure to 30 nM thapsigargin was less effective and induced weaker nuclear accumulation (Fig. 3a, b). By contrast, stimulation with 100 nM thapsigargin in high $K^+$ solution led to significantly more NFAT nuclear accumulation (Fig. 3a, b).

The histogram in Fig. 3c compares nuclear accumulation of NFAT for individual resting (non-stimulated) cells and for cells exposed to 2 μM thapsigargin for 40 min. The majority of resting cells showed a nuclear/cytosolic ratio ≪1 and the entire population exhibited relatively little variance. Cells stimulated with 2 μM thapsigargin all showed a clear increase in nuclear accumulation and the population response was right-shifted several fold (Fig. 3c). A similar analysis comparing NFAT movement in 30 nM thapsigargin in standard $K^+$ solution with 100 nM thapsigargin in high $K^+$ solution showed a striking difference in pattern (Fig. 3d). Many cells failed to show resolvable NFAT nuclear migration to 30 nM thapsigargin (Fig. 3d), with nuclear/cytosolic ratios similar to resting cells (Fig. 3c). Interestingly, those cells that did respond to 30 nM thapsigargin showed a clear increase in nuclear/cytosolic ratio (Fig. 3d). By contrast, most cells responded to 100 nM thapsigargin in high $K^+$ solution with a migration profile (Fig. 3d) that was qualitatively similar to 2 μM thapsigargin (Fig. 3c).

The distribution of NFAT responses to 30 nM thapsigargin was reminiscent of a bimodal distribution, with some cells not responding at all and others responding strongly. We used a binomial distribution to describe the spread in the data at different nuclear proportions (Supplementary Fig. 3). Binomial fits to the data showed that the nuclear/cytosolic ratio in resting cells exhibited an approximately normal distribution (Fig. 3e, upper panel). After stimulation with 2 μM thapsigargin, all cells clustered around large responses (Fig. 3e). Responses to either 100 nM thapsigargin or 100 nM thapsigargin in high $K^+$ solution were very similar to those evoked by 2 μM thapsigargin (Fig. 3e). However, the responses to 30 nM thapsigargin deconstructed nicely into a bimodal distribution; ~20% of the cells failed to respond but the fraction that did respond did so in a manner largely indistinguishable from responses seen in higher thapsigargin concentrations (Fig. 3e). The weak aggregate NFAT1 responses to 30 nM thapsigargin (Fig. 3b) therefore reflect contributions from both responding and non-responding cells rather than weak responses across all cells. One explanation for a bimodal pattern of NFAT activation would be that $Ca^{2+}$ influx itself is bimodal, with some cells failing to respond to 30 nM thapsigargin and others responding by generating maximal $Ca^{2+}$ entry. However, $Ca^{2+}$ influx rate did not exhibit a bimodal pattern for any of the thapsigargin concentrations tested (Supplementary Fig. 4).

These data show that 30 nM thapsigargin in normal $K^+$ solution is considerably less effective in activating NFAT1 than

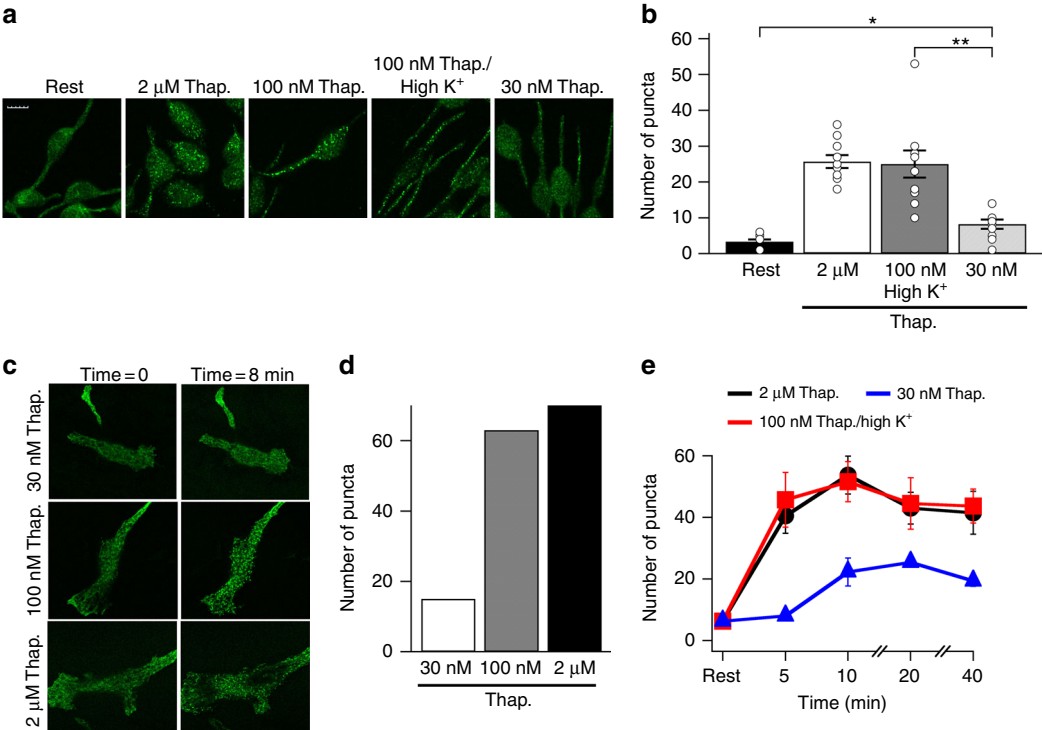

**Fig. 2** Increasing thapsigargin concentration recruits more STIM1 puncta. **a** Endogenous STIM1 puncta are compared for the conditions shown. Cells were fixed 10 min after stimulation. Rest denotes non-stimulated cells that were kept in standard solution for 10 min prior to fixing. Whole scale bar is 10 μm. **b** Aggregate data from experiments as in panel **a** are compared. Each bar denotes > 10 cells. **c** Snapshots from live imaging experiments are shown from cells expressing STIM1-YFP. Time = 0 denotes STIM distribution just prior to thapsigargin application. Time = 8 min denotes images taken 8 min after thapsigargin exposure. **d** Aggregate data are compared. **e** The number of STIM1-YFP puncta are compared over time following stimulation with thapsigargin at the concentrations indicated. Each point is the mean of > 10 cells. *denotes $p < 0.05$ and **$p < 0.01$, determined using unpaired Student's $t$-test. Error bars denote SEM

100 nM thapsigargin in high $K^+$ solution, despite both stimuli increasing bulk cytosolic $Ca^{2+}$ to similar levels.

**Dependence of c-fos expression on thapsigargin concentration**. We measured c-fos expression using quantitative PCR (qPCR), stimulating cells with thapsigargin for the same duration as used for NFAT migration. Although 100 nM thapsigargin evoked a significant increase in c-fos mRNA (Fig. 4a), very small increases above resting levels were seen with either 30 nM thapsigargin in standard $K^+$ solution or 100 nM thapsigargin in high $K^+$ solution (Fig. 4a). We also measured c-fos protein expression at a cellular level using immunocytochemistry[16]. 100 nM thapsigargin triggered a significant increase in c-fos protein expression above resting levels (Fig. 4b), but neither 30 nM thapsigargin in standard $K^+$ solution nor 100 nM thapsigargin in high $K^+$ solution evoked a significant increase (Fig. 4b). Activation of c-fos expression and NFAT1 migration (averaged for all cells i.e., responders and non-responders) are compared in Fig. 4c, for the different stimuli shown on the abscissa, applied for the same time. Thirty nanomolar thapsigargin did not activate c-fos and caused a modest but significant increase in NFAT1 migration. One-hundred nanomolar thapsigargin in high $K^+$ solution was a very weak stimulus for c-fos, but was much more effective in stimulating NFAT (Fig. 4c). Although c-fos and NFAT activation pathways are both stimulated by $Ca^{2+}$ microdomains near CRAC channels, they nevertheless have different requirements for the number of STIM1 puncta. They also exhibit different sensitivities to $Ca^{2+}$; in response to the same trigger (namely 100 nM thapsigargin in high $K^+$ solution), only NFAT was activated.

**Comparison of NFAT and c-fos gene expression**. The preceding data suggest that NFAT1 is robustly activated by 100 nM thapsigargin in high $K^+$ solution, whereas c-fos is not. As we measured NFAT1 and c-fos activities using different techniques (fluorescence microscopy versus qPCR), one explanation for our results could reflect a possible difference in sensitivity between the two methods. To address this, we compared c-fos and NFAT activities using the same experimental techniques.

First, we compared c-fos and NFAT-driven gene expression to 100 nM thapsigargin in high $K^+$ solution with 30 nM thapsigargin in standard $K^+$ solution using qPCR. In RBL cells, $Ca^{2+}$ entry through CRAC channels increases interleukin-5 (IL-5) expression through a pathway absolutely dependent on NFAT activity[27], although the increase in IL-5 mRNA is modest. Pretreatment with the phorbol ester phorbol 12-myristate 13-acetate (PMA) increases $Ca^{2+}$-dependent IL-5 expression considerably[28]. To increase the bandwidth of IL-5 detection, we stimulated cells with thapsigargin following exposure to phorbol ester. Phorbol ester alone failed to increase IL-5 levels (Fig. 4d; labelled Rest). Application of either 2 μM or 100 nM thapsigargin resulted in a substantial increase. Stimulation with 30 nM thapsigargin failed to increase IL-5 expression, whereas a significant increase was seen with 100 nM thapsigargin in high $K^+$ solution (Fig. 4d). We repeated the c-fos experiments under identical conditions of PMA exposure. PMA alone failed to affect c-fos expression (Fig. 4e; labelled Rest). Similarly, neither 30 nM thapsigargin in standard $K^+$ solution nor 100 nM thapsigargin in high $K^+$ solution increased c-fos expression (Fig. 4e), consistent with the data in Fig. 4a. However, robust increase in c-fos expression was seen to 100 nM thapsigargin (Fig. 4e). Therefore, using qPCR, c-

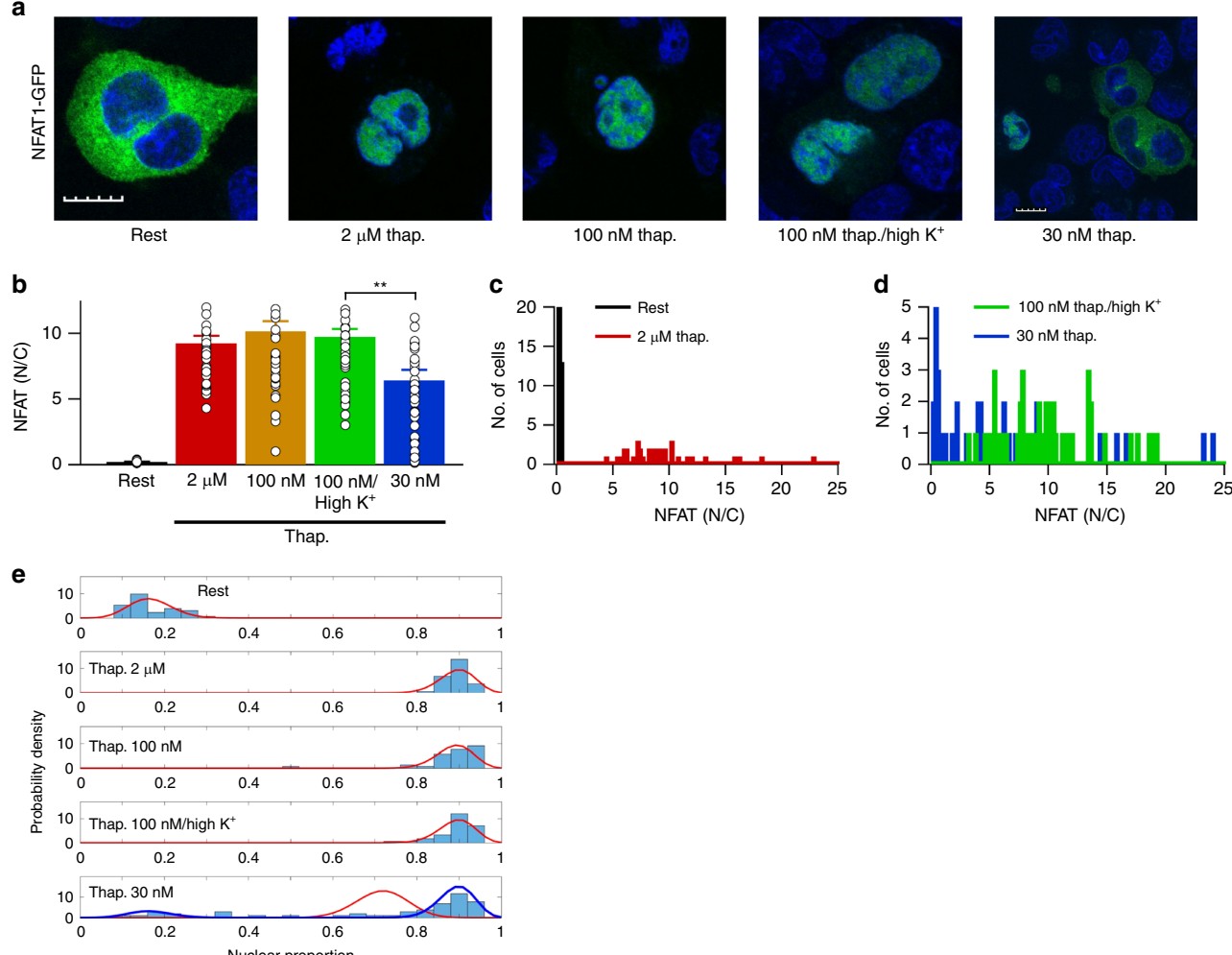

**Fig. 3** Effect of different concentrations of thapsigargin on NFAT1 nuclear accumulation. **a** Confocal images compare NFAT1-GFP distribution between cytosol and nucleus for the conditions shown. Cells were stimulated for 40 min. Whole scale bar is 10 μm. Nuclei were stained with DAPI (blue). **b** Aggregate data from experiments as in panel **a** are compared. Each bar is > 30 cells. **c** Histogram compares nuclear accumulation between resting (non-stimulated) cells and cells challenged with 2 μM thapsigargin. **d** Histogram compares NFAT1-GFP distribution between cells challenged with 30 nM thapsigargin in standard $K^+$ solution and 100 nM thapsigargin in high $K^+$ solution. **e** Binomial distributions of nuclear/cytosolic NFAT1-GFP are derived from data as in panels **c** and **d**. The red trace for 30 nM thapsigargin (representing the data) could be deconstructed into two distributions (blue), reflecting non-responders and responders. **denotes $p < 0.01$, determined using unpaired Student's $t$-test. Error bars denote SEM

fos and NFAT-dependent responses are differentially activated by 100 nM thapsigargin in high $K^+$ solution. Stimulation with 100 nM thapsigargin in either standard or high $K^+$ solution increased NFAT1 translocation to the nucleus (Fig. 3b) and subsequent IL-5 gene transcription (Fig. 4d). However, although challenge with 30 nM thapsigargin caused a modest but significant increase in NFAT migration (Fig. 3b), this failed to increase IL-5 expression significantly (Fig. 4d; $p = 0.09$). We surmise that this reflects, at least in part, that qPCR measurements of IL-5 transcription reflect the combination of responders and non-responders in the population, as well as slower NFAT nuclear dynamics to 30 nM thapsigargin[16].

In a second approach, we expressed the transcription factor STAT5 tagged with GFP and measured its distribution between the cytosol and nucleus before and after stimulation using fluorescence microscopy. Exposure to 2 μM or 100 nM thapsigargin increased nuclear accumulation of STAT5-GFP (Supplementary Fig. 5). However, as was the case with c-fos expression, challenge with either 30 nM thapsigargin or 100 nM thapsigargin in high $K^+$ solution failed to increase nuclear levels of STAT5-GFP, compared with non-stimulated cells (Supplementary Fig. 5).

**FACS analysis of c-fos and NFAT-reporter gene expression.** In a third approach, we compared c-fos and NFAT1 expression in the same cells using fluorescence-activated cell sorting (FACS). We co-transfected YFP (under a c-fos promoter) and RFP (under an NFAT promoter). Little expression of either protein occurred under resting conditions (Fig. 5a), when compared with non-transfected cells (labelled Blank in Fig. 5a). The histograms in Fig. 5b, c show YFP and RFP expression, respectively, for the conditions indicated. Stimulation with 30 nM thapsigargin evoked little increase in YFP expression compared with resting cells (Fig. 5a, b) but caused a significant increase in RFP levels (Fig. 5a, c). Stimulation with 100 nM thapsigargin in high $K^+$ solution also failed to increase c-fos expression (Fig. 5a, b) but led to significantly more expression of RFP than seen with 30 nM thapsigargin (Fig. 5a, c). The median fluorescence value from all cells for each condition is stated in the histograms, as is the % of cells that exhibited a fluorescence intensity value > $10^3$, a value selected because this was twofold larger than the largest intensity seen in the blank group. Compared with non-stimulated resting cells, stimulation with 30 nM thapsigargin or 100 nM thapsigargin in high $K^+$ solution led to only very small increases in the

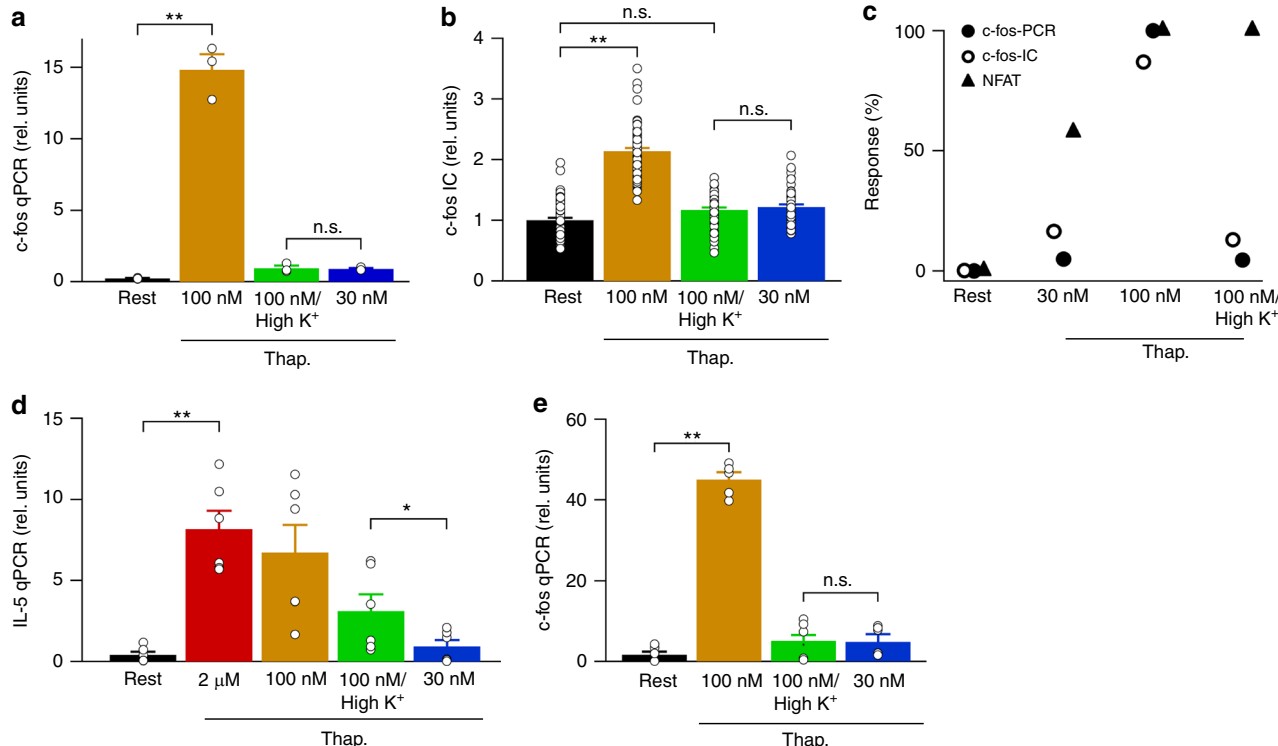

**Fig. 4** C-Fos expression requires strong store depletion. **a** c-fos was measured using qPCR for the conditions shown. Cells were stimulated with thapsigargin for 40 min before mRNA was isolated. Data are mean of three independent experiments. **b** c-fos protein was quantified using immunocytochemistry. Each bar denotes > 56 cells. **c** Graph compares normalised c-fos and NFAT responses to the stimuli shown. C-Fos measurements using qPCR and immunocytochemistry (IC) are both shown. NFAT response was the nuclear/cytosolic ratio. Responses were normalised to those obtained after stimulation with 2 μM thapsigargin. **d** Bar chart compares IL-5 transcription for the different conditions indicated. Rest denotes non-stimulated cells. Data are mean of four independent experiments. **e** Bar chart compares c-fos transcription for the conditions shown. Data are mean of three independent experiments. In panels **d** and **e**, cells were stimulated with thapsigargin and PMA (50 ng/ml). Rest denotes transcription in non-thapsigargin-treated cells but which had been exposed to PMA. *denotes $p < 0.05$, **$p < 0.01$ and n.s. not significant, determined using unpaired Student's $t$-test. Error bars denote SEM

% of cells expressing YFP and in the median fluorescence intensity of YFP (Fig. 5b). By contrast, both stimuli led to a large increase in % cells expressing RFP and to a very substantial increase in the median values (Fig. 5c). Whereas the % of cells expressing YFP increased from 0.5% to just 2% following stimulation with 30 nM thapsigargin, the corresponding increase for RFP in the same cells was from 0.5 to 54%.

These data confirm that 30 nM thapsigargin in standard $K^+$ solution activates NFAT-dependent gene expression more robustly than c-fos in the same cells and that 100 nM thapsigargin in high $K^+$ solution is a stronger stimulus for NFAT activation than 30 nM thapsigargin in standard $K^+$ solution. In these experiments, we measured protein expression ~12 h after stimulation. We considered the possibility that c-fos expression might have already declined considerably by this time point. We therefore measured c-fos-YFP and NFAT-dependent RFP protein expression over different times. Both proteins had increased significantly by 6 h (Fig. 5d, e). Importantly, YFP remained relatively stable up to 24 h and RFP expression increased gradually over this time period. The fraction of cells expressing the proteins was similar over this time period.

The data in Fig. 5c are compatible with a bimodal distribution of NFAT-RFP expression to thapsigargin stimulation. As the FACS studies provided data from > 15,000 cells per condition, we assessed bimodality in this large population (Fig. 5f). We plotted log fluorescence intensity against cell number. The histograms were very nicely fitted (red trace) as the sum of two normal distributions (non-responders and full responders). For the Rest

group, the data were well-represented by a single non-responder normal distribution (100% of cells). For cells stimulated with 30 nM thapsigargin, the experimental data could be represented by a distribution (shown in red) that was the sum of two normal distributions (shown in blue) representing the non-responders (accounting for ~40% of the population) and the responders (~60% of the population). Corresponding distributions for cells challenged with 100 nM thapsigargin in high $K^+$ solution were ~30% non-responders. For responding cells, the mean RFP fluorescence was similar for cells challenged with 30 nM thapsigargin and 100 nM thapsigargin in high $K^+$ solution (Fig. 5f).

It is interesting that stimulation with 30 nM thapsigargin increased NFAT-RFP expression significantly, whereas the same stimulus was less effective for IL-5 transcription. This probably reflects the fact that transiently transfected plasmids are generally not inserted into the genome and exhibit an increased transcriptional efficiency.

**Ca$^{2+}$-dependence of NFAT and c-fos**. The results shown in Fig. 4c (100 nM thapsigargin in high $K^+$ solution group) suggest that the NFAT activation pathway has a higher sensitivity to local $Ca^{2+}$ than c-fos, despite both being dependent on $Ca^{2+}$ microdomains near CRAC channels. A prediction would, therefore, be that NFAT should be activated more strongly than c-fos in the presence of lower external $Ca^{2+}$ concentrations, since the latter condition produces $Ca^{2+}$ microdomains of smaller size[29]. We

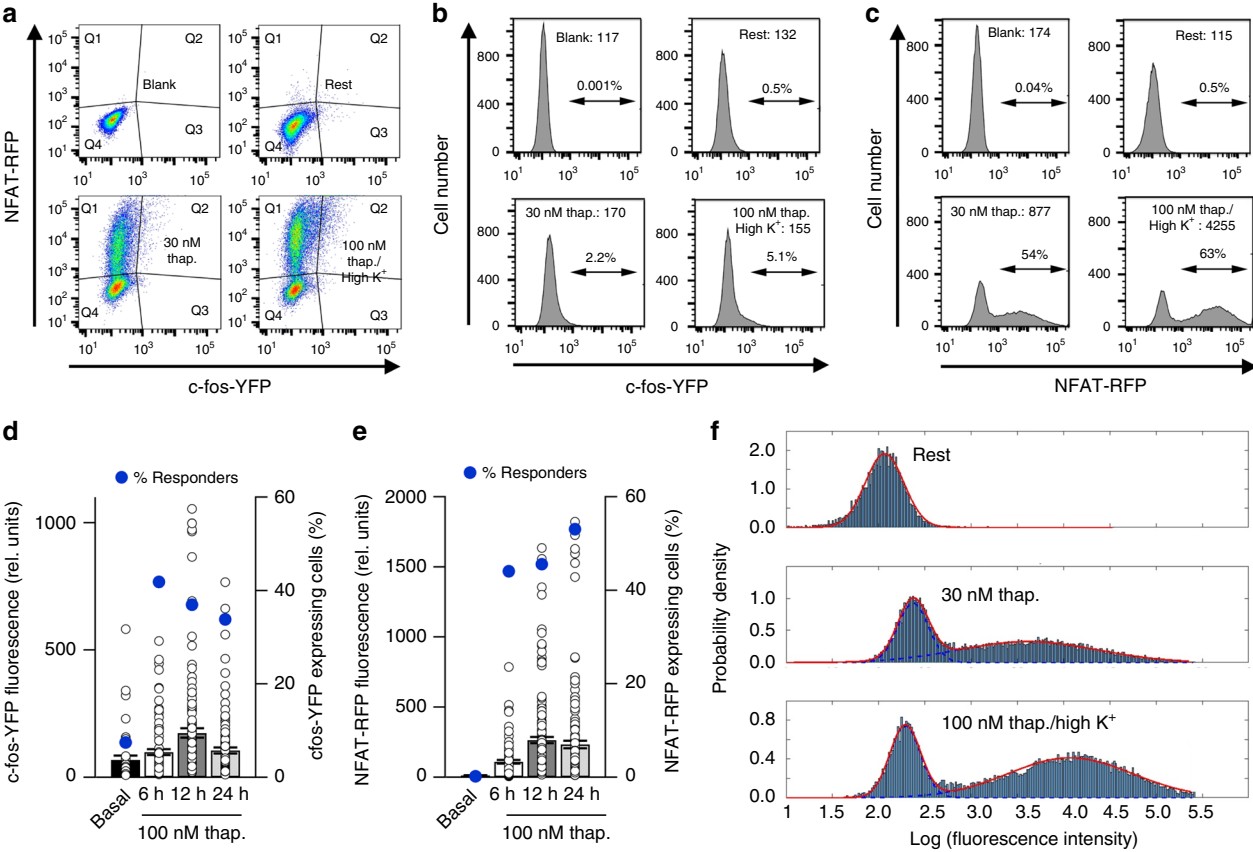

**Fig. 5** Comparison of extent of co-expression of c-fos-YFP and NFAT-RFP in the same cells. **a** Representative flow cytometry plots of c-fos-driven YFP and NFAT-driven RFP expression in non-transfected cells (labelled Blank), transfected but unstimulated cells (labelled Rest), cells challenged with 30 nM thapsigargin and cells stimulated with 100 nM thapsigargin in high $K^+$ solution. Plots were divided into four sections (Q1–Q4), representing high RFP expression (Q1), high RFP and YFP expression (Q2), high YFP expression (Q3) and low expression of both proteins (Q4). For blank, % cells in Q1 was 0.044, Q2 was 0.009, Q3 was 0.005 and Q4 was 99.9. For rest, % cells in Q1 was 0.46, Q2 was 0.1, Q3 was 0.63 and Q4 was 98.7. For 30 nM thapsigargin, % cells in Q1 was 50.9, Q2 was 2.16, Q3 was 0.32 and Q4 was 46.6. For 100 nM thapsigargin in high $K^+$ solution, % cells in Q1 was 60.4, Q2 was 5.1, Q3 was 0.23 and Q4 was 34.2. **b** Histograms plot the fluorescence of YFP for each condition. Median values are stated in each histogram and the percentages denote the % of cells exhibiting a fluorescence intensity of $\geq 10^3$. **c** Histograms plot the fluorescence of RFP for the conditions shown. Median and % values have the same meaning as in panel **b**. **d** Time-course of c-fos-eYFP fluorescence is compared at different times after stimulation. The fraction of cells expressing c-fos at different times is compared (denoted % responders) as is the fluorescence intensity of these individual responding cells (43–133 cells for each condition). **e** Time-course of NFAT-driven RFP fluorescence is shown at different times after stimulation (1–182 cells for each time point; only 1 out of 579 non-stimulated (basal) cells were NFAT-positive). For d and e, expression was monitored using epifluorescence microscopy. **f** Histograms, derived from the FACS data, with fitted distributions shown in red for non-responders and responders as described in the methods. Blue dashed lines indicate the two normal distributions that are summed to form bimodal cases

applied different external $Ca^{2+}$ concentrations (0.1–2 mM) to cells pre-stimulated with 2 µM thapsigargin in $Ca^{2+}$-free solution (Fig. 6a). The rate of rise of cytosolic $Ca^{2+}$ increased as external $Ca^{2+}$ increased (Fig. 6b). The bulk cytosolic $Ca^{2+}$ rise declined relatively rapidly in 0.1 mM $Ca^{2+}$ (Fig. 6a), indicating that the plasma membrane $Ca^{2+}$ ATPase pumps were able to reduce cytosolic $Ca^{2+}$ effectively. Compared with non-stimulated cells (denoted Rest), confocal images showed that nuclear accumulation of NFAT1-GFP increased following stimulation with 2 µM thapsigargin in 0.1 mM external $Ca^{2+}$ and this became considerably stronger in 0.5 mM $Ca^{2+}$ and 2 mM $Ca^{2+}$ (Fig. 6c; aggregate data in Fig. 6d). C-Fos expression, measured using qPCR, increased modestly in 0.5 mM external $Ca^{2+}$ but was undetectable in 0.1 mM $Ca^{2+}$ (Fig. 6e). Hence the NFAT activation pathway has a higher sensitivity to local $Ca^{2+}$ than the pathway used for c-fos. To test this more directly, we compared c-fos and IL-5 expression in cells from the same preparations following stimulation with thapsigargin in different external $Ca^{2+}$

concentrations. Whereas IL-5 transcription was similar in the presence of 0.5 or 2 mM $Ca^{2+}$ (Fig. 6f, upper panel), significantly less c-fos transcription occurred in 0.5 mM $Ca^{2+}$ (Fig. 6f, lower panel).

**Effect of raising bulk $Ca^{2+}$ on NFAT1 and c-fos activation.** One possibility for why NFAT is activated more robustly than c-fos at lowered external $Ca^{2+}$ is that it exhibits a higher sensitivity for $Ca^{2+}$ than c-fos and, therefore, requires fewer STIM1-Orai1 puncta and/or less $Ca^{2+}$ flux through each punctum. To address this, we induced a large rise in bulk $Ca^{2+}$ uniformly throughout the cell, independent of CRAC channels. Stimulation of RBL cells with thapsigargin/$0Ca^{2+}$/$La^{3+}$ solution results in a large and prolonged rise in cytosolic $Ca^{2+}$ as released $Ca^{2+}$ can no longer be exported out of the cell by the $La^{3+}$-sensitive plasma membrane $Ca^{2+}$ATPase pump[13]. The $Ca^{2+}$ rise under these conditions should be uniform and not produce spatial gradients below the plasma membrane. Therefore, responses should be dictated

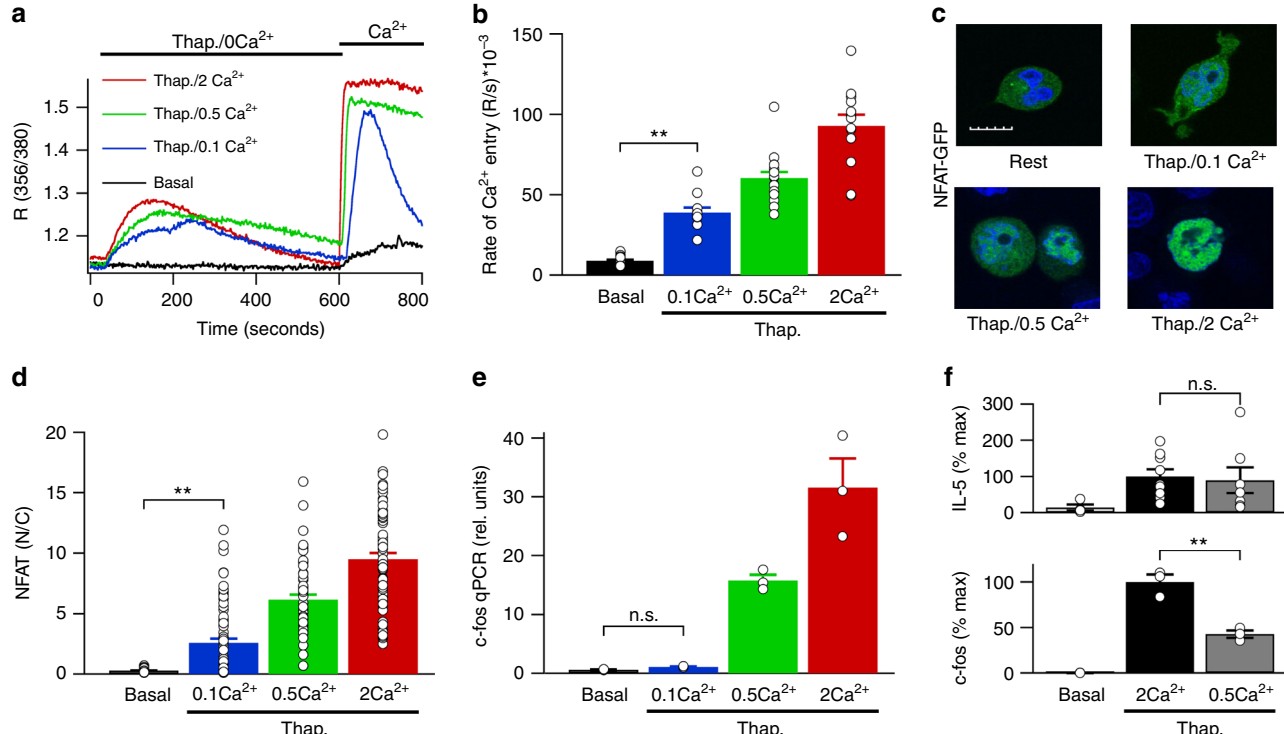

**Fig. 6** Effects of varying external $Ca^{2+}$ on NFAT translocation and gene transcription. **a** Store-operated $Ca^{2+}$ entry was measured following readmission of different external $Ca^{2+}$ concentrations, following challenge with 2 μM thapsigargin in $Ca^{2+}$-free solution. **b** Aggregate data from experiments as in panel **a** are compared. Each bar denotes 12–17 cells. **c** Confocal images compare NFAT1-GFP distribution following stimulation with thapsigargin in the presence of different external $Ca^{2+}$ concentrations. Whole scale bar denotes 10 μm. Nuclei were stained blue with DAPI. **d** Aggregate data from experiments as in panel **c** are compared. Each bar is the mean of between 56 and 70 cells. **e** C-Fos expression, measured using qPCR, is shown for the different conditions. **f** Transcription of IL-5 (upper panel) and c-fos (lower panel) are compared between non-stimulated resting cells (basal) and after stimulation with thapsigargin (2 μM) in either 2 mM or 0.5 mM external $Ca^{2+}$. Cells from the same preparations were used for IL-5 and c-fos measurements. Both groups were treated with PMA. Data have been normalised to the response to thapsigargin in 2 mM external $Ca^{2+}$. **denotes $p < 0.01$ and n.s. not significant, determined using unpaired Student's $t$-test. Error bars denote SEM

by their relative $Ca^{2+}$ affinities. One-hundred nanomolar thapsigargin/$0Ca^{2+}$/$La^{3+}$ stimulation increased c-fos expression only to approximately 20% that evoked by 100 nM thapsigargin in 2 mM $Ca^{2+}$-containing external solution (Fig. 7a). $La^{3+}$ does not impair $Ca^{2+}$-dependent c-fos transcription in RBL cells[30]. By contrast, stimulation with 100 nM thapsigargin/$0Ca^{2+}$/$La^{3+}$ enhanced nuclear accumulation of NFAT1 significantly (Fig. 7b; the response was 36% that induced by 100 nM thapsigargin in 2 mM $Ca^{2+}$-containing external solution). Histograms are shown for the NFAT1 nuclear/cytosolic ratio in unstimulated resting cells (Fig. 7c), for cells stimulated with 100 nM thapsigargin in 2 mM $Ca^{2+}$ (Fig. 7d) and for cells challenged with 100 nM thapsigargin/$0Ca^{2+}$/$La^{3+}$ (Fig. 7e). The mean values are listed in each histogram. The histograms in Fig. 7f represent application of the binomial model to data obtained from experiments carried out in panels 7c–e. Stimulation with thapsigargin in either 2 mM $Ca^{2+}$ or $0Ca^{2+}$/$La^{3+}$ solution were both reasonably well accounted for by a single binomial distribution with no non-responders.

The finding that stimulation with thapsigargin/$0Ca^{2+}$/$La^{3+}$ solution elicited a lower average nuclear/cytosolic ratio for NFAT accumulation than thapsigargin in 2 mM $Ca^{2+}$ could be explained by $La^{3+}$ entering the cytosol and exerting a partial inhibitory effect on a step in NFAT activation. However, application of ionomycin (5 μM) in external $Ca^{2+}$ after cells had been challenged with thapsigargin/$0Ca^{2+}$/$La^{3+}$ increased NFAT translocation (Supplementary Fig. 6), ruling out an inhibitory effect of $La^{3+}$ on NFAT activation.

**Agonist dose–response curves to NFAT1 and c-fos activation.** Based on the preceding data, we reasoned that lower concentrations of agonist, which induce fewer $Ca^{2+}$ microdomains, should be more effective in stimulating NFAT than c-fos. To test this, we activated endogenous Gq-coupled cysteinyl leukotriene type I receptors with the agonist leukotriene $C_4$ ($LTC_4$). Stimulation with sub-maximal concentrations of $LTC_4$ elicited a series of cytosolic $Ca^{2+}$ oscillations, which run down in $Ca^{2+}$-free solution (Fig. 8a). Readmission of external $Ca^{2+}$ resulted in a rise in cytosolic $Ca^{2+}$ due to store-operated $Ca^{2+}$ entry Fig. 8a[30]). The relationship between agonist concentration and the rate of $Ca^{2+}$ entry, shown in Fig. 8b, could be fitted with a Hill-type equation yielding an $EC_{50}$ of 10 nM and a Hill coefficient of ~1.

$LTC_4$ evoked dose-dependent nuclear accumulation of NFAT1-GFP (Fig. 8c). Increasing agonist concentration increased the % of cells that exhibited NFAT1 movement (Fig. 8d). For all cells that responded over the concentration range tested, NFAT nuclear accumulation was similar (Fig. 8e), as reported previously[31]. The dose–response relationship revealed an $EC_{50}$ of 1 nM for NFAT1 activation (Fig. 8f). Although the relationship between $LTC_4$ concentration and c-fos expression (measured using qPCR) was also dose-dependent (Fig. 8f), it was right-shifted, with an $EC_{50}$ of 10 nM (Fig. 8f).

We also measured expression of an NFAT-regulated GFP reporter gene and compared this with c-fos protein expression in the same cells and to the same concentration of agonist. Stimulation with $LTC_4$ led to a dose-dependent increase in both

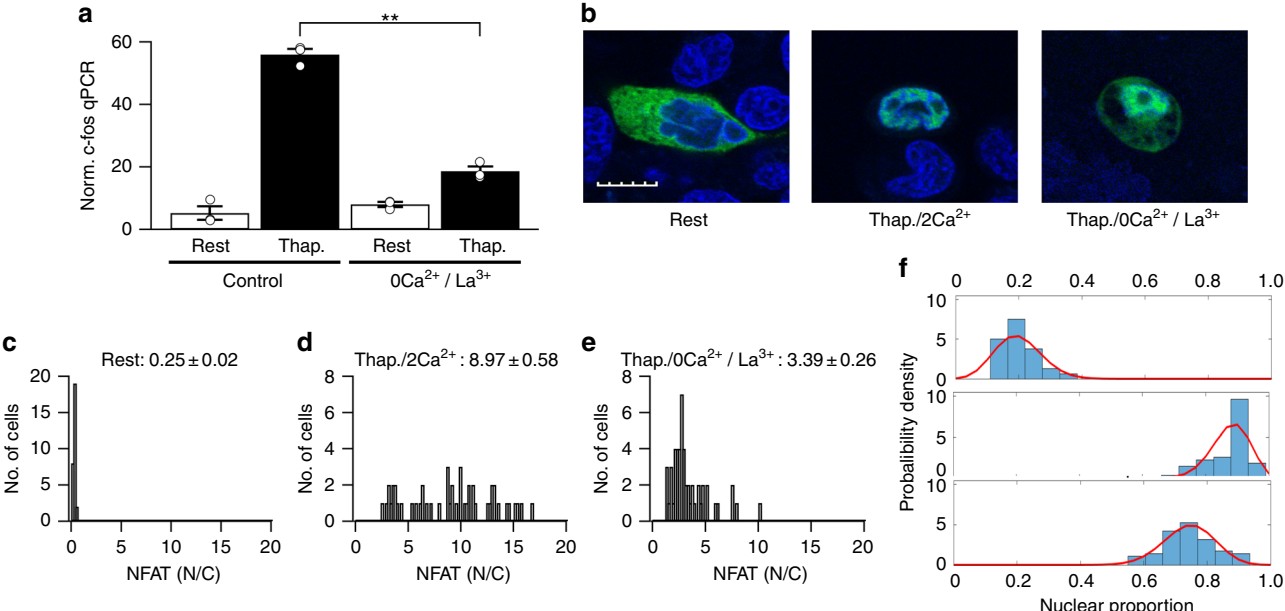

**Fig. 7** Effects of a uniform rise in bulk cytosolic $Ca^{2+}$ on activation of NFAT and c-fos. **a** C-Fos transcription, measured with qPCR, is compared for the conditions indicated. Data are mean of three independent experiments. **b** NFAT1-GFP nuclear migration is compared for the conditions shown. Images were obtained using confocal microscopy. Whole scale bar is 10 μm. **c** Histogram plots the NFAT1-GFP nuclear/cytosolic ratio for resting (unstimulated) cells. **d** As in panel **c** but cells were stimulated with thapsigargin in 2 mM $Ca^{2+}$. **e** As in panel **c** but the stimulus was thapsigargin in $Ca^{2+}$-free solution supplemented with 1 mM $La^{3+}$. The mean nuclear/cytosolic values for all cells in each condition are shown above each histogram. **f** Binomial distribution fits to the data from panels **c–e** are compared. Thapsigargin was used at 2 μM in all panels. **denotes $p < 0.01$, determined using unpaired Student's $t$-test. Error bars denote SEM

c-fos and GFP expression. However, the relationship between NFAT-regulated reporter gene expression and agonist concentration was left-shifted compared with the corresponding curve for c-fos (Fig. 8g). Therefore, under identical conditions, LTC4 driven NFAT-dependent gene expression occurs at lower agonist concentrations than c-fos expression.

## Discussion

$Ca^{2+}$ microdomains near STIM1-Orai1 $Ca^{2+}$ channel complexes activate the transcription factors c-fos and NFAT. Although transcription of some genes is regulated by NFAT and c-fos acting in combination, others are activated only by NFAT[32]. This raises a paradox: if $Ca^{2+}$ microdomains near open CRAC channels activate both transcription factors, how can NFAT be recruited independently of c-fos? More generally, if two pathways can be stimulated by the same local $Ca^{2+}$ signal, can one be selectively recruited? Our data help resolve this by revealing that the transcription factors have different sensitivities to local $Ca^{2+}$. NFAT exhibits higher sensitivity and is selectively recruited at lower levels of stimulus intensity. Co-operativity between NFAT and c-fos in transcriptional control therefore will depend on agonist concentration. Low levels of receptor stimulation will favour NFAT activation but, as stimulus intensity increases, c-fos would be recruited additionally. Differences in transcription factor sensitivity to $Ca^{2+}$ increase the bandwidth of gene expression programmes through a combination of independent and co-operative interactions between NFAT and c-fos.

NFAT and c-fos also have distinct requirements on the number of STIM1-Orai1 puncta formed. Whereas NFAT1 activated to some extent when only a fraction of the total number of puncta that could form did so, there was no increase in c-fos. C-Fos activity requires Syk-dependent phosphorylation of STAT5[15]. Although Syk is associated with Orai1, it remains so after store depletion and therefore is confined to the plasma membrane[15,16]. The likelihood

of cytosolic STAT5 reaching a small fraction of STIM1-Orai1 puncta will be low. However, the probability that Syk will encounter and thereby activate STAT5 will rise with an increase in puncta number. By contrast, a pool of NFAT1 and its activator, calcineurin, are already associated with the plasma membrane at rest through binding to AKAP79 and are then brought to the realm of the $Ca^{2+}$ microdomain through interaction between the N-terminus of Orai1 and AKAP79[19]. This, coupled with the high sensitivity of the NFAT pathway to local $Ca^{2+}$ ensures activation in the presence of fewer STIM1-Orai1 puncta.

Although it is well established that NFAT migrates into the nucleus after CRAC channel activation, we have found that variability in translocation can be faithfully represented by a binomial distribution. With a sub-maximal stimulus (30 nM thapsigargin), NFAT migration distributed into either non-responders or responders, with the latter exhibiting responses that were quantitatively similar to those induced by maximally effective stimuli. The population average showed nuclear/cytosolic ratios that were intermediate between those seen in unstimulated cells and in maximally stimulated cells, but this reflected the ratio of non- and maximal responders rather than a graded range of responding cells. Store-operated $Ca^{2+}$ entry induced by 30 nM thapsigargin was relatively similar between cells and no evidence for bimodal $Ca^{2+}$ entry was found. Therefore, bimodal NFAT activation in the presence of external $Ca^{2+}$ is an intrinsic property of the downstream NFAT pathway itself. It is instructive to explore how a bimodal distribution of NFAT1 activation might occur mechanistically. An elegant study by Okamura et al.[33] demonstrated a conformation switch mechanism for NFAT1 activation. At rest, NFAT1 was maintained in an inactive state by phosphorylation of fourteen conserved serine residues. $Ca^{2+}$-dependent dephosphorylation of thirteen of these sites by calcineurin exposed a nuclear localisation signal and masking of a nuclear export signal, enabling NFAT to migrate to and remain within the nucleus. Salazar and Hoefer[34] have developed an

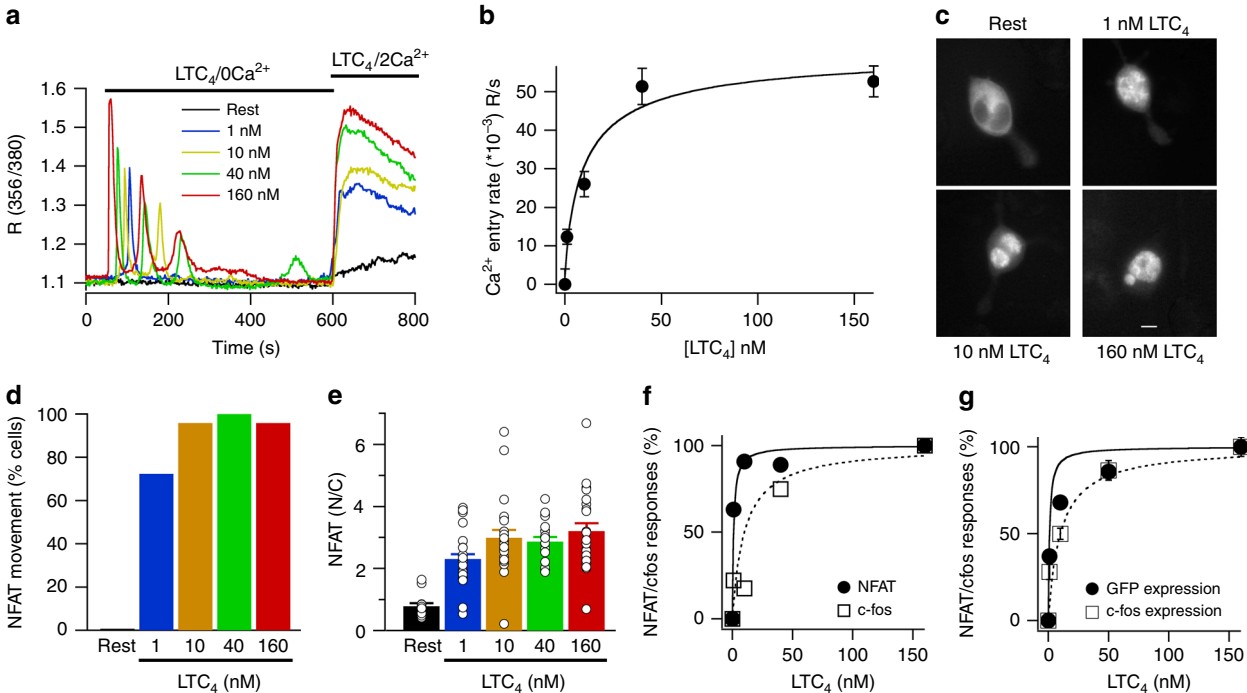

**Fig. 8** Relationship between $LTC_4$ concentration and activation of NFAT and c-fos. **a** Cytosolic $Ca^{2+}$ signals to different concentrations of $LTC_4$ are compared. Cells were stimulated with $LTC_4$ in $Ca^{2+}$-free solution and then external $Ca^{2+}$ was readmitted as shown. **b** Graph plots the rate of $Ca^{2+}$ entry (from panel **a**) versus $LTC_4$ concentration. Each point is the mean of 15–30 cells. **c** Images compare distribution of NFAT1-GFP for the conditions shown. Images were taken 40 min after stimulation. Rest image was taken after 40 min in the same external solution but without stimulus. **d** The % of cells that showed nuclear migration of NFAT1 is compared for different agonist concentrations. **e** Bar chart compares NFAT1-GFP nuclear/cytosolic ratio following challenge with different $LTC_4$ concentrations. Each bar is 14–29 cells. **f** NFAT1 translocation and c-fos transcription (qPCR) are compared following stimulation with different $LTC_4$ concentrations. **g** Graph compares co-expression of c-fos protein and GFP under an NFAT promoter in the same cells. Cells were stimulated with different concentrations of $LTC_4$ for 40 min and then agonist was washed out. Cells were placed in fresh medium in the incubator for 12 h and then fixed. C-fos expression was measured using immunocytochemistry and NFAT-reporter gene expression by GFP fluorescence. C-fos responses reflect the extent of protein expression normalised to the response evoked by 160 nM $LTC_4$. NFAT responses are represented by the % of GFP-positive cells, normalised to the response evoked by 160 nM $LTC_4$. This parameter was used because the extent of reporter gene expression is independent of $LTC_4$ stimulus intensity[31]. $LTC_4$ $EC_{50}$ values were 1 nM for GFP expression and 9.8 nM for c-fos expression. Dose–response curves were fitted to a modified Hill equation: %Response $= [LTC_4]^n / ([LTC_4]^n + (EC_{50})^n)$, where $n$ denotes Hill coefficient

insightful quantitative model that explains the conformational switch mechanism. NFAT activation could be represented mathematically as a binary response and that multiple phosphorylation sites resulted in a threshold for NFAT stimulation, where almost complete dephosphorylation was required for full protein activation. Weak calcineurin stimulation led to modest NFAT dephosphorylation and therefore minor activation. By contrast, strong calcineurin stimulation resulted in dephosphorylation of the requisite thirteen phosphoserine sites followed by full activation of NFAT1. Our data can be rationalized in this conceptual framework. AKAP79 brings calcineurin and NFAT1 close to the CRAC channel pore[19], enabling strong activation of the enzyme by high local $Ca^{2+}$. Therefore, only a few calcineurin molecules will need to be recruited to the $Ca^{2+}$ microdomain in order for effective dephosphorylation of proximal NFAT1, especially as rephosphorylation of NFAT1 within the cytosol is relatively slow[27]. With few STIM1-Orai1 puncta forming in response to a modest stimulus such as 30 nM thapsigargin, the probability of AKAP79, with calcineurin and NFAT1 in tow, associating with a sufficient number of active Orai1 channels will be low and therefore only a fraction of the total pool of available calcineurin will be recruited. Nevertheless, as calcineurin will be activated robustly by the high local $Ca^{2+}$ and one stimulated calcineurin enzyme can dephosphorylate multiple NFAT1 proteins, this will result in non-linear activation of NFAT. Additionally, store-

operated $Ca^{2+}$ entry stimulates migration of calcineurin into the nucleus[35], enabling active enzyme both to access cytosolic phosphorylated NFAT, as well as reduce nuclear export of the transcription factor. The kinetics of binding of the AKAP79/calcineurin/NFAT1 complex to the small number of active Orai1 channels that form in response to weak stimulation will be slow. Consistent with this, migration of NFAT1 to the nucleus is considerably slower with low levels of stimulation, but nevertheless reaches the same extent of nuclear accumulation as a high-intensity stimulus[16].

NFAT activation was unimodal following stimulation with high concentrations of thapsigargin, reflecting strong responses from all cells in the population. A somewhat milder response was obtained in response to challenge with thapsigargin in $Ca^{2+}$-free solution containing $La^{3+}$. Although the distribution of NFAT between nucleus and cytosol in thapsigargin/$0Ca^{2+}$/$La^{3+}$ was qualitatively similar to that evoked by maximal thapsigargin concentration, the mean response was ~50% of the latter despite being unimodal. One possible explanation is that the bulk $Ca^{2+}$ rise induced by thapsigargin/$0Ca^{2+}$/$La^{3+}$ raises local $Ca^{2+}$ uniformly in the vicinity of AKAP79/Orai1 but to a lesser extent than that seen when $Ca^{2+}$ microdomains near CRAC channels are activated by thapsigargin in the presence of external $Ca^{2+}$.

Genetically identical *E. coli* switch between different phenotypes in a stochastic manner. For those bacteria with the lac

operon, a switch from a bimodal to a unimodal distribution has been found in a manner dependent on inducer intensity[36]. Our results demonstrating a switch from bimodal to unimodal distribution depending on stimulus strength extend the findings of Choi et al.[36] to a mammalian system.

Our results show that transcription factors respond to different numbers of STIM1-Orai1 puncta and, for similar numbers of puncta, different transcription factors require different amounts of $Ca^{2+}$ flux through the channels, as dictated by their relative $Ca^{2+}$ sensitivities. Variations in puncta number are determined by stimulus intensity. Flux through Orai1 is governed by channel pore properties that define unitary conductance and $Ca^{2+}$ selectivity, and the prevailing electrical gradient for $Ca^{2+}$ entry. $Ca^{2+}$ selectivity of Orai1 is not an immutable feature of the channels but is tuned by STIM1 binding[37], Single channel optical recordings using a $Ca^{2+}$ indicator tethered to the Orai1 channel have revealed multiple channel open states, imparting variable $Ca^{2+}$ flux through individual channels[38]. The electrical driving force for $Ca^{2+}$ entry will depend on the activities of other open ion channels that set the membrane potential. Large fluctuations in membrane potential through $Ca^{2+}$-dependent activation of non-selective cation channels have been described in T cells[39] and pancreatic acini[40], providing a mechanism to dynamically regulate $Ca^{2+}$ flux through Orai1. Therefore, multiple regulatory mechanisms converging on the number of STIM1-Orai1 puncta that form, as well as $Ca^{2+}$ flux through the channels enable the same agonist to couple $Ca^{2+}$ microdomains to selective downstream signalling pathways.

## Methods

**Cell culture**. RBL-2H3 cells were purchased from ATCC (via UK supplier LGC) and were cultured at 37 °C with 5% $CO_2$ in Dulbecco's modified Eagle's medium (DMEM) supplemented with 10% fetal bovine serum and 1% penicillin/streptomycin. Cells were split using Trypsin-EDTA and plated onto glass coverslips for use 24–48 h later.

**Plasmids and transfection**. STIM1-YFP (gift from Dr. Tobias Meyer, Stanford), NFAT1-GFP (gift from Dr. Jennings Morley, NIH) and both the EGFP-based and RFP-based reporter plasmids containing an NFAT promoter (gifts from Dr. Yuri Usachev, Iowa) were transfected into RBL-2H3 cells using the Amaxa system. pOTTC589-pAAV c-fos Nuc-eYFP (nuclear localised eYFP under the c-fos promoter) was obtained from Addgene, deposited by Dr. Brandon Harvey.

**Fluorescence $Ca^{2+}$ measurements**. Cytosolic $Ca^{2+}$ measurements were carried out at room temperature using the IMAGO charged-coupled device camera-based system from TiLL photonics, as described[16]. Cells were loaded with Fura-2/AM in the dark for 40 min, washed and left for 15 min for further de-esterification. Cells were excited at 356 and 380 nm (20 ms exposures at 0.5 Hz) and emission was collected > 505 nm. External solution was composed of 145 mM NaCl, 2.8 mM KCl, 2 mM $CaCl_2$, 2 mM $MgCl_2$, 10 mM D-glucose, 10 mM HEPES, pH 7.4 with NaOH. $Ca^{2+}$-free solution was composed of 145 mM NaCl, 2.8 mM KCl, 2 mM $MgCl_2$, 10 mM D-glucose, 10 mM HEPES, 0.1 mM EGTA, pH 7.4 with NaOH. High $K^+$-containing external solution was composed of 100 mM KCl, 45 mM NaCl, 2 mM $CaCl_2$, 2 mM $MgCl_2$, 10 mM HEPES, 10 mM D-glucose, pH 7.4 with KOH. $Ca^{2+}$ signals are plotted as R, which denotes the 356/380 nm ratio.

**NFAT1-GFP translocation**. Twenty-four hours after expression of NFAT1 tagged with GFP, cells were stimulated with either thapsigargin or $LTC_4$ for 40 min and then fixed in 4% paraformaldehyde at room temperature. Resting (control) cells were treated identically but received no stimulus. Images were obtained using an Olympus FV1000 confocal microscope and nuclear and cytosolic distribution, obtained by drawing regions of interest of identical size, were analysed using Image J. Nuclei were co-stained with DAPI. In some experiments (Fig. 8), NFAT1-GFP translocation was measured using an Olympus xCellence system. Nuclear and cytosolic distributions were obtained by drawing regions of interest of identical size and then analysed using Image J.

**Gene reporter assay**. Following transfection of RFP-based reporter plasmid under an NFAT promoter, cells were stimulated with $LTC_4$ for 40 min in the incubator as described[31]. Cells were then washed several times with DMEM and left in the incubator for 12 h. After that, the % of cells expressing RFP fluorescence was

quantified. Gene expression was defined as fluorescence 3xSD > cell auto-fluorescence, measured at 485 nm excitation.

**Immunocytochemistry**. After appropriate stimulation (see text), coverslips were washed twice in ice cold PBS and then fixed with 4% paraformaldehyde for 15 min at room temperature and permeabilized with PBS/5% BSA/0.3% Triton. Cells were stained with anti-STIM1 antibody (1:100, BD Transduction Laboratories™) or anti-c-fos antibody (1:100, Santa Cruz Biotechnology) overnight at 4 °C. Coverslips were then washed three times in PBS and incubated with anti-mouse Alexa 488 secondary antibody (ThermoFisher Scientific) for STIM1 (1:200) or anti-rabbit Alexa Fluor 568 (1:200) antibody (Thermo Fisher Scientific) for 1.5 h at room temperature. Coverslips were then washed in PBS, then preserved in mounting medium (Vector Laboratories). Images were taken with an Olympus FV1000 confocal microscope.

**Fluorescence-activated cell sorting**. Cells were co-transfected with pNFAT-TA-mRFP (for NFAT-driven RFP expression) and pOTTC589- pAAV c-fos Nuc-eYFP using the Amaxa system. Twenty-four hours after transfection, cells were treated with different concentrations of thapsigargin (as stated in the text) for 40 min and then washed several times in medium without stimulus. After 12 h incubation, cells were fixed with 4% paraformaldehyde and washed three times with PBS. The YFP and RFP expression in individual cells were detected using a BD X-20 flow cytometer. Data were analysed using FlowJo software.

**Real-time quantitative RT-PCR**. For measurement of c-fos transcription, RBL-2H3 cells were stimulated with thapsigargin or $LTC_4$ for 40 min (to match the measured NFAT1-GFP translocation time) in external $Ca^{2+}$-containing solution. RNA was then extracted as described below. For measurement of IL-5 and c-fos from the same cell preparations, cells were pretreated with PMA (50 ng/ml) for 5 min, exposed to thapsigargin and PMA for 40 min and then washed and maintained in fresh medium for a further 50 min. Thereafter, RNA was extracted using an RNeasy Mini Kit (Qiagen), as described[16]. RNA was quantified spectrophotometrically by absorbance at 260 nm. Total RNA (1 µg) was reverse-transcribed using the iScriptTM cDNA synthesis kit (Bio-Rad), according to the manufacturer's instructions. Quantitative real-time RT-PCR was performed with cDNA, Taqman Universal PCR Master Mix (Applied Biosystems), $H_2O$ and specific primers for Taqman Gene expression assays (Rn00487426_g1 for rat c-fos; Rn01459975_m1 for rat IL-5 and Rn00667869_m1 for rat actin). The samples were loaded into 96-well plates and analysed by the ABI Prism 7000 Sequence Detection System software (Applied Biosystems). The qPCR conditions were as follows: 2 min at 50 °C, 10 min at 95 °C, followed by 40 cycles of 15 s at 95 °C and 1 min at 60 °C. For quantification, the relative quantities of samples were calculated according the comparative $\Delta C_t$ method and normalized to β-actin.

**Whole-cell patch clamp recordings**. Patch clamp experiments were conducted in the tight seal whole-cell configuration at room temperature (20–24 °C) as previously described[21]. Pipettes, pulled from borosilicate glass, Sylgard-coated and then fire-polished had resistances of 3–6 MOhms when filled with internal solution containing: 145 $Cs^+$-glutamate, 8 mM NaCl, 2 mM $MgCl_2$, 10 mM HEPES, 2 mM Mg-ATP, 0.3 mM EGTA, 2 mM pyruvic acid, 2 mM K-malate, 1 mM $KH_2PO_4$, pH 7.2 (CsOH). Pyruvic acid, malate and $KH_2PO_4$ were included to ensure mitochondria remained energised and therefore were able to maintain $Ca^{2+}$ buffering in whole-cell recording[41]. External solution contained 145 mM NaCl, 2.8 mM KCl, 2 mM $CaCl_2$, 2 mM $MgCl_2$, 10 mM CsCl, 10 mM HEPES, 10 mM D-glucose, pH 7.4 with NaOH. The CRAC current was measured by applying voltage ramps (−100 to + 100 mV in 50 ms) at 0.5 Hz from a holding potential of 0 mV. Currents were filtered using an 8-pole Bessel filter at 2.5 kHz and digitised at 100 µs. Capacitive currents were compensated before each ramp or step by using the automatic compensation of the EPC 9–2 amplifier. Leak currents were subtracted by averaging 3–5 ramp currents obtained shortly after break-in.

Membrane potential was measured using the current clamp mode. Pipettes were pulled from borosilicate glass, Sylgard-coated and fire-polished. Pipette resistances were 4–5 Mohms when filled with a pipette solution containing 145 $K^+$-glutamate, 8 mM NaCl, 2 mM $MgCl_2$, 10 mM HEPES, 2 mM Mg-ATP, 0.1 mM EGTA, pH 7.2 (KOH). External solution contained 145 mM NaCl, 2.8 mM KCl, 2 mM $CaCl_2$, 2 mM $MgCl_2$, 10 mM HEPES, 10 mM D-glucose, pH 7.4 with NaOH. High $K^+$-containing external solution contained 100 mM KCl, 45 mM NaCl, 2 mM $CaCl_2$, 2 mM $MgCl_2$, 10 mM HEPES, 10 mM D-glucose, pH 7.4 with KOH. Membrane potential was measured over the first few seconds after break-in.

**Distributions**. Viewing fluorescence data in terms of nuclear:cytoplasmic ratio is a standard approach, but we found that when attempting to fit distributions to characterise such data that a log-Normal distribution fitted some unimodal histograms, and a Normal distribution fitted others. To circumvent this, we transformed the data into 'Nuclear Proportion' (in the range zero to one). We then found that a binomial distribution fitted all the unimodal histograms well (Fig. 3e, g). This distribution assumes a certain number of agents make a probabilistic decision whether or not to allow NFAT1 into the nucleus: i.e., data = Binomial(n, p); with the same value of 'n' (number of agents) and different values of 'p' (probability of nuclear NFAT). In our

setting, rather than numbers of molecules, we hypothesise that a number of 'NFAT activating units' are involved (which could be one punctum, or all the puncta in a region of the cell, or downstream AKAP79/calcineurin that becomes the bottleneck). In a low 30 nM dose of thapsigargin, a single binomial distribution was not a good fit (red lines in the fourth panels of Fig. 3e, g), and the data are consistent with a proportion of NFAT activating units being active and others remaining at rest.

For modelling of the FACS data (Fig. 5), for the Rest group, the logarithmically transformed fluorescence data were well-represented by a single non-responder normal distribution, i.e.,

$$\log_{10}(H) \sim \mathcal{N}(\mu, \sigma) \tag{1}$$

where '$H$' denotes the absolute fluorescence, '$\mathcal{N}$' indicates a normal distribution, the Rest data gave a mean of $\mu = 2.06$ and a standard deviation of $\sigma = 0.21$.

For cells stimulated with thapsigargin, the experimental data were well described by fitting two normal distributions, as shown in Fig. 3:

$$\log_{10}(H) \sim \alpha \mathcal{N}(\mu_n, \sigma_n) + (1 - \alpha)\mathcal{N}(\mu_r, \sigma_r) \tag{2}$$

where $\alpha$ represents the proportion of non-responding cells, the subscript '$n$' represents the non-responders, and '$r$' represents the responders. With 30 nM thapsigargin the non-responders accounted for 38% of the population ($\alpha = 0.38$) and the responders 62%. The fluorescence distributions of non-responders were given by $\mu_n = 2.37$, $\sigma_n = 0.17$ and responders by $\mu_r = 3.61$, $\sigma_r = 0.75$. Corresponding distributions for cells challenged with 100 nM thapsigargin in high K$^+$-containing external solution were 31% non-responders ($\alpha = 0.31$) with a fluorescence distribution of $\mu_n = 2.29$, $\sigma_n = 0.17$ and 69% responders with a distribution of $\mu_r = 4.07$, $\sigma_r = 0.67$.

The log of the data was normally distributed, but in original fluorescence units the data were very skewed. The reason the means of FACS data were somewhat counter-intuitively above the means obtained from the fitted normal distributions was because of this skew. This was reflected in the difference between mean and median values from each group. The mean of FACS Rest group was 137.9 and the median was 114.2, in good agreement. However, the mean of FACS 30 nM thapsigargin group was 8994.9, whereas the median was 857.0, and corresponding values for the 100 nM thapsigargin/high K$^+$-containing external solution group were 19214.0 (mean) and 4420.2 (median).

**Statistics**. Data are expressed as means ± SEM. No specific randomization or blinding protocols were used. Different groups were compared using a two-tailed paired or unpaired Student $t$-test. In all graphs, * and ** denotes $p$-values < 0.05, 0.01, respectively and n.s. denotes not significant.

## Data availability

Data supporting the findings of this manuscript are available from the corresponding author upon reasonable request. A reporting summary for this Article is available as a Supplementary Information file. The source data underlying Figs. 1, 2, 3, 4, 5, 6, 7, 8 and Supplementary Figs. 1, 4, 5, 6 are provided as a Source Data file.

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

## Acknowledgements

This work was supported by an MRC Programme grant (L01047X) to A.B.P. and a Sir Henry Dale Fellowship jointly funded by the Wellcome Trust and Royal Society (grant number 101222/Z/13/Z) to G.R.M.

## Author contributions

Y.-P.L., D.B. and A.B.P. designed, analysed and interpreted experiments. G.R.M. and A.B. P. developed the model and G.R.M. modelled the data. All authors contributed to the writing of the manuscript.

## Additional information

**Competing interests:** A.B.P. is Scientific Founder of Calcico Therapeutics. The remaining authors declare no competing interests.

