## [Peer Review File · Nature Communications]

Reviewers' Comments:

Reviewer #1:

Remarks to the Author:

This is a very interesting paper dealing with the mechanisms underlying the selective activation of Ca²⁺-dependent transcription factors evoked by Ca²⁺ store-depletion induced formation of STIM1-Orai1 clusters that results in Ca²⁺ influx.

At the heart of this study is a very logical and careful quantitative analysis of the relationship between the degree of store depletion and STIM1-Orai1 cluster formation and therefore Ca²⁺ influx as well as careful comparison of transcription factor translocation at varying levels of store depletion and at varying levels of Ca²⁺ inflow through open CRAC channels. To the best of my knowledge such a fundamental study has never been carried out before. As the Ca²⁺ flux through open channels depends on the electrochemical gradient for Ca²⁺ across the plasma membrane, the flux can be altered by changing the membrane potential. In practice, as shown in this paper, the number of STIM1-Orai clusters can be varied by changing the thapsigargin concentration and the flux through individual CRAC channels can be varied by changing the external K⁺ concentration.

This study shows convincingly how Ca²⁺-dependent activation of NFAT and cFOS can be selective. These two transcription factors have different affinities for Ca²⁺, with NFAT being the more sensitive and therefore able to be recruited at low levels of stimulus intensity. However, this is only one aspect. This study also shows that NFAT and cFOS have different requirements for the number of STIM1-Orai puncta formed. This also plays a significant role for selective activation.

Overall, this study provides much new, valuable and, importantly, quantitative information about transcription factor activation and is therefore a fundamentally important work, which will be of great and general interest to the whole signal-transduction field. This paper should therefore be published in Nature Communications.

I have only two rather minor critical points:

[1] The paper generally provides all relevant information about the results described, with the exception of Fig. 4, where we are not given information about the numbers represented by the columns.

[2] It would have been helpful if the figures had been numbered. The reference on l. 246 and 247 to Figs 2E and 2F is confusing as Fig. 2 only seems to have segments A-D.

Reviewer #2:

Remarks to the Author:

Lin Yu-Ping et al. Selective activation of Ca²⁺-dependent transcription factors through variations in the numbers of, and Ca²⁺ influx through, STIM1-Orai1 clusters.

In this manuscript, authors try to address whether different activation levels of store-operated Ca²⁺ entry can mediate activation of distinct Ca²⁺-dependent transcription factors. The authors show that two Ca²⁺-dependent transcription factors, NFAT1 and c-fos, have different requirements for the number of STIM1-Orai1 clusters and on the Ca²⁺ flux through these clusters. NFAT activation required fewer clusters and was more robustly activated than c-fos by low concentrations of agonist. Moreover, even for the same number of clusters, recruitment of transcription factors occurred sequentially, due to differences in their Ca²⁺ affinities. The concept of different Ca²⁺-requirement for activation of the NFAT and c-fos is novel, but the techniques used and data interpretation in this manuscript raise many concerns. The title itself is ambiguously written and the exact meaning of "Ca²⁺ flux" is not clear. My specific comments are:

1. The techniques used as readout of two Ca^{2+} dependent activation of two pathways are vastly different in terms of sensitivity and accuracy. For example, nuclear translocation of GFP-NFAT is a very direct read out of NFAT activation and appropriate to check activity of this pathway. However, activation of c-fos is not directly Ca^{2+} -dependent, it indirectly depends on Ca^{2+} via activation of STAT5. In addition, use of qPCR and immunocytochemistry for c-fos cannot serve as quantitative outputs. If the authors want to use qPCR as read out of pathway activation, they should examine expression of gene regulated by NFAT, so that qPCR can be used as readout of activation of both the pathways. In my understanding this is a major flaw and we are not comparing “apples to apples” by using different techniques as read outs of NFAT or c-fos pathways. Ideal comparison would be checking expression of an NFAT-dependent gene and comparing it to that of c-fos to draw the conclusions of the manuscript.

2. Low TG treatment also means that not all SERCA pumps are inhibited. However, in the experimental design in figure 1, there is no consideration for this aspect and the proposed model is too simplified. Partial block of SERCA can lead to faster decrease of cytoplasmic Ca^{2+} and refilling of ER Ca^{2+} stores. Thus, it can influence the stability of the STIM1 puncta. Also, I wonder if the method of blocking store-operated Ca^{2+} entry by K^{+} can be the best way to decrease Ca^{2+} influx due to its broad effects. This model needs to be tested with more specific methods using pharmacological or genetic methods. Also, the authors need to measure CRAC currents to show similar CRAC currents in 30 nM TG v/s 100 nM TG with K^{+} to provide unequivocal evidence of similar CRAC currents.

3. In Figure 1D-G, obviously, the response of each cells varies, especially when treated with low concentration of thapsigargin. The population of NFAT translocation has been quantified in this figure, but the method used in analysis of Ca^{2+} entry is not clearly written. Which cells are selected and what is the criteria? In general, the methods are not described in detail at all.

4. Figures 2E and F are missing. Also, in Figures 2A-D, it is not clearly shown that only the number of STIM1 puncta are influenced by low concentration of TG treatment as the authors claim. This claim is important for the main hypothesis; hence accurate kinetics should be shown.

5. The same goes for Figures 5 and 6. However, if we assume the techniques are accurate, these data are the major findings in this manuscript by showing clear differences between the NFAT and c-fos pathways.

In addition, this manuscript is lacking proper statistical analyses – for example in Fig. 3B, the authors say 2-fold difference between 20 nM and 100 nM plus high K^{+} , however, there is not statistics to show that there is significant difference between the two bars. Since there are major concerns regarding the primary experimental design used over and over again to claim that NFAT and c-fos pathways require different Ca^{2+} -levels for activation, I believe, this manuscript in its current form is not suitable for publication in Nature Communications. The authors will have to compare expression of an NFAT-dependent gene to that of c-fos using qPCR and immunocytochemistry to compare the kinetics and sensitivity of activation of these two pathways.

Reviewer #3:

Remarks to the Author:

These interesting studies report of selective response of Ca^{2+} -regulated transcription factors to Ca^{2+} microdomains formed by STIM1-Orai1 clustering at membrane contact site. The authors compared the effect of the same Ca^{2+} influx through small and large number of formed STIM1-Orai1 micro-clusters on activation of NFAT1 and c-fos. Moreover, the authors continue to show that the number of STIM1-Orai1 clusters, rather than simply the cytoplasmic concentration of Ca^{2+} , selectively determines the range and robustness of Ca^{2+} -dependent activation of the transcription factor NFAT1.

The studies are thorough and of high quality and the authors provide convincing evidence to support their conclusions. I have only few minor comments that should be addressed prior to publication.

The authors relay on inducing variable store depletion to control the number of STIM1-Orai1 puncta. Puncta number and Ca²⁺ influx are analyzed after 10 min treatment with thapsigargin. However, activation of transcription factors is analyzed 40 min after thapsigargin treatment. The authors should assay Ca²⁺ influx and puncta formation 40 min after thapsigargin treatment to ensure that the difference in Ca²⁺ influx and puncta formation is retained when transcription factors activation is measured.

The results in Figure 3e,g are quite impressive and important. Should resolution permit, the authors should attempt similar binomial analysis of STIM1 puncta in Figure 2. This may provide further support for the relationship between STIM1 puncta numbers and NFAT1 activation.

A control experiment should test the effect of loading the cells with BAPTA on STIM1-Orai1 clustering and the number of STIM1 puncta

Please provide statistics throughout in the Figures or legend (Figures 1g, i, 3b, e, g, 4b, 5b, h, 6c, e, j, 7d, e).

Introduction ref 16: Some caution should be indicated in relation to claiming CRAC channels involvement in pancreatitis in view of the study reporting severe bacteremia and death by inhibition of acinar Orai1 (see PMID: 28273482).

Line 143: add of after each.

Line 291: no arrow is included in 3e.

We thank the reviewers for their thoughtful and constructive comments on our manuscript, which we have found very helpful. We have addressed all the comments made and have conducted several new experiments that confirm and extend our original observations.

Reviewer 1.

We thank the reviewer for his/her very positive and supportive comments. This referee found the work novel and important and concluded “Overall, this study provides much new, valuable and, importantly, quantitative information about transcription factor activation and is therefore a fundamentally important work, which will be of great and general interest to the whole signal-transduction field. This paper should therefore be published in Nature Communications”.

The reviewer had only two minor comments.

1. The paper generally provides all relevant information about the results described, with the exception of Fig. 4, where we are not given information about the numbers represented by the columns.

We apologise for this omission. The relevant information has now been included in all Figure Legends.

2. It would have been helpful if the figures had been numbered. The reference on l. 246 and 247 to Figs 2E and 2F is confusing as Fig. 2 only seems to have segments A-D.

We apologize for not numbering the Figures. We have now corrected this oversight. We have corrected lines 246/247.

Reviewer 2.

We thank the reviewer for his kind and helpful comments. The reviewer raised several points, which we have addressed by carrying out numerous new experiments. We believe the new data strengthen and extend our original findings. We are grateful to the reviewer for his/her detailed critique and for the insightful comments.

1. The techniques used as readout of two Ca²⁺ dependent activation of two pathways are vastly different in terms of sensitivity and accuracy. For example, nuclear translocation of GFP-NFAT is a very direct read out of NFAT activation and appropriate to check activity of this pathway. However, activation of c-fos is not directly Ca²⁺-dependent, it indirectly depends on Ca²⁺ via activation of STAT5. In addition, use of qPCR and immunocytochemistry for c-fos cannot serve as quantitative outputs. If the authors want to use qPCR as read out of pathway activation, they should examine expression of gene regulated by NFAT, so that qPCR can be used as readout of activation of both the pathways. In my understanding this is a major flaw and we are not comparing “apples to apples” by using different techniques as read outs of NFAT or c-fos pathways. Ideal comparison would be checking expression of an NFAT-dependent gene and comparing it to that of c-fos to draw the conclusions of the manuscript.

Thank you for this comment. The referee makes an important point and we have carried out several new experiments that address this concern directly. The new data are included as the new Figures 4D and 4E, the entire new Figure 5, the new Figure 6F and the new Supplemental Figure 5.

Specifically, we have carried out the following experiments.

- 1) We have compared expression of c-fos with that of an NFAT-dependent gene (interleukin 5; Kar, Mirams, Christian and Parekh *Molecular Cell* 64, 746-759) using quantitative real time PCR. High concentrations of thapsigargin increased transcription of both genes. However, stimulation with 100 nM thapsigargin in high K⁺-containing external solution failed to increase c-fos expression (new Figure 4E) but caused a significant increase in interleukin 5 (new Figure 4D). These new data, using the same technique of qPCR, have been added to the manuscript.
- 2) As suggested by reviewer 4, we have studied an upstream readout of the c-fos pathway, namely Ca²⁺-dependent activation of the transcription factor STAT5. Our previous work showed that local Ca²⁺ entry through CRAC channels activated the non-receptor tyrosine kinase Syk, tethered close to the CRAC channel (Samanta et al., *Cell Reports* 12, 203-216), which then phosphorylated STAT5 (Ng et al, *JBC* 284, 24767-24772). Once phosphorylated, STAT5 dimerises and migrates to the nucleus where it activates c-fos transcription (Darnell, *Science* 277, 1630-1635). We expressed STAT5-GFP and monitored its translocation from the cytosol to the nucleus following CRAC channel activation (Supplemental Figure 5). The new data demonstrate robust STAT5 migration to the nucleus following stimulation with high doses of thapsigargin (2 μ M or 100 nM). However, stimulation with either 30 nM thapsigargin or 100 nM thapsigargin in high K⁺-containing external solution failed to induce STAT5 movement compared with resting cells. This is in excellent agreement with the c-fos transcription data, where neither 30 nM thapsigargin nor 100 nM thapsigargin in high K⁺-containing external solution increases c-fos expression (Figure 4E). By contrast, we find strong nuclear accumulation of NFAT-GFP in response to 100 nM thapsigargin in high K⁺-containing external solution and modest but significant accumulation to 30 nM thapsigargin (Figure 3). Therefore, using fluorescence microscopy with GFP-tagged NFAT1 or GFP-tagged STAT5, we demonstrate that STAT5 is not activated under conditions that lead to strong stimulation of NFAT1. The STAT5 translocation data have been included as the new Supplemental Figure 5.
- 3) We have co-expressed RFP under an NFAT promoter together with a nuclear localized eYFP construct under a c-fos promoter. We have used FACS to quantify expression of RFP and eYFP in the same cells to the same stimulus. These data are included as the entire new Figure 5. The data show that neither 30 nM thapsigargin in normal K⁺-containing external solution nor 100 nM thapsigargin in high K⁺-containing external solution increase eYFP expression (indicating low c-fos activity) whereas RFP expression is high (indicating enhanced NFAT activity) in the same cells. We believe that this set of experiments, carried out in the same cells, demonstrate unequivocally that the different transcription factors require different numbers of STIM1-Orai1 puncta as well as different extents of Ca²⁺ flux through the same number of puncta. The FACS studies provide data from >15,000 cells per condition. We

have applied our new binomial spread function to these data and find they fit the empirical results very well. This new analysis is also included in Figure 5.

2. Low TG treatment also means that not all SERCA pumps are inhibited. However, in the experimental design in figure 1, there is no consideration for this aspect and the proposed model is too simplified. Partial block of SERCA can lead to faster decrease of cytoplasmic Ca²⁺ and refilling of ER Ca²⁺ stores. Thus, it can influence the stability of the STIM1 puncta. Also, I wonder if the method of blocking store-operated Ca²⁺ entry by K⁺ can be the best way to decrease Ca²⁺ influx due to its broad effects. This model needs to be tested with more specific methods using pharmacological or genetic methods. Also, the authors need to measure CRAC currents to show similar CRAC currents in 30 nM TG v/s 100 nM TG with K⁺ to provide unequivocal evidence of similar CRAC currents.

We have directly addressed the question the reviewer asks, namely do stores refill in the presence of a low thapsigargin concentration. We find that stores do not refill in low thapsigargin in the continuous presence of external Ca²⁺. The data have been included as the new Supplemental Figure 1 and this issue is discussed in detail on page 5.

In addition to using high K⁺, we have tried the additional approaches suggested by the reviewer. Knockdown of Orai1 using an siRNA-based strategy reduces both c-fos and NFAT responses (e.g. Kar et al., PNAS 109, 6969-6974). Similarly, pharmacological block of CRAC channels (Ng et al., JBC 283, 31348-3135) inhibited c-fos and NFAT activation. However, we have found that higher concentrations of the inhibitor BTP2 are required to suppress NFAT activation than c-fos (a concentration close to the IC₅₀ of 2 μM inhibits c-fos by 76±7% and NFAT by 41±3%; 2 μM thapsigargin was the stimulus). This would be consistent with c-fos having lower sensitivity to Ca²⁺ and therefore requiring a greater number of CRAC channels to open. However, we are hesitant to draw any firm conclusions from these experiments because we do not know how drug interact with the CRAC channels at a molecular level. It is possible that BTP2 fully blocks some STIM1-Orai1 channel clusters completely, leaving others intact. Alternatively, the drug could block a fraction of the channels in each punctum. The local Ca²⁺ will be very different between these two scenarios, complicating interpretation. We believe a detailed understanding of how the drugs interact with the channels, which has not been undertaken before and which is necessary for rigorous interpretation of our data, is beyond the scope of the present study.

The patch clamp experiment the reviewer suggests is not possible to do. When we patch clamp in high K⁺-containing external solution, the inwardly rectifying K⁺ current in RBL cells is very large, typically of the order of several hundred pA (see Straube and Parekh, Pfluegers Archiv 444, 389-396). This large current masks the small I_{CRAC} that develops (typically 50 pA). Therefore, to record the CRAC current, we routinely have to block the inward rectifier. This would reduce the extent of depolarisation to high K⁺ because the inward rectifier provides the main resting K⁺ conductance in these cells. Moreover, in all the experiments in the current manuscript, the inward rectifier is active and therefore it would be difficult to compare the two sets of data (active versus blocked inward rectifier).

3. In Figure 1D-G, obviously, the response of each cells varies, especially when treated with low concentration of thapsigargin. The population of NFAT translocation has been quantified in this figure, but the method used in analysis of Ca^{2+} entry is not clearly written. Which cells are selected and what is the criteria? In general, the methods are not described in detail at all.

We apologise for not making this clearer. We quantified the rate of Ca^{2+} entry by differentiating the Ca^{2+} signal that was generated upon readmission of external Ca^{2+} . This is a standard approach in the field and has been widely used by us and others. As we stated in the text (page 12-13), we have not observed a large Ca^{2+} entry rate to 30 nM thapsigargin in some cells and a very low rate in others. Unlike the NFAT distribution, the calcium signals are not binomial. However, we thank the reviewer for raising this interesting point. We have constructed histograms to the four main conditions we have used (basal Ca^{2+} entry, Ca^{2+} entry to 30 nM thapsigargin, Ca^{2+} entry to 100 nM thapsigargin and Ca^{2+} entry to 100 nM thapsigargin in high K^+ -containing external solution). Compared with basal Ca^{2+} entry, the histograms show that all cells respond to thapsigargin stimulation, although they generate different Ca^{2+} entry rates as expected from differences in store depletion. However, the data show that Ca^{2+} influx rate is not bimodal to any of the thapsigargin concentrations tested, in sharp contrast to NFAT. We have included these data as the new Supplemental Figure 4 and discuss the data towards the end of page 7.

4. Figures 2E and F are missing. Also, in Figures 2A-D, it is not clearly shown that only the number of STIM1 puncta are influenced by low concentration of TG treatment as the authors claim. This claim is important for the main hypothesis; hence accurate kinetics should be shown.

We apologise for reference to the absent Figures 2E and 2F. In light of comments from referee 3 (see point 1 to referee 3), we have carried out new experiments to measure the kinetics of puncta formation and stability. The new data are included as Figure 2C. The data show that i) STIM1 puncta are stable over our 40 minutes recording period and ii) 30 nM thapsigargin induces fewer puncta than 100 nM thapsigargin in high K^+ -containing external solution over the time course of our experiments.

5. The same goes for Figures 5 and 6, compare c-fos and IL-5 by PCR

Please see detailed response above (point 1). Note that the original Figures 5 and 6 are now Figures 6 and 7. To address the reviewer's point, we have compared c-fos and IL-5 transcription to thapsigargin stimulation in the presence of either 2mM or 0.5 mM external Ca^{2+} . Whereas IL-5 expression is similar under these conditions, the c-fos response is significantly smaller in 0.5 mM Ca^{2+} (new Figure 6F). Therefore, using the same technique of qPCR, our new data confirm differences in Ca^{2+} sensitivity between NFAT and c-fos driven responses. We have removed the BAPTA data from the original Figure 6, because this did not allow us to pinpoint distances between CRAC channels and the transcription factors and therefore added little to the manuscript. We have focussed instead on the $0Ca^{2+}/La^{3+}$ experiments.

In addition, this manuscript is lacking proper statistical analyses – for example in Fig. 3B, the authors say 2-fold difference between 20 nM and 100 nM plus high K+, however, there is not statistics to show that there is significant difference between the two bars.

Thank you for pointing this out. We have now added statistical analyses to all relevant Figures.

Reviewer 3.

We thank the reviewer for his/her very kind and positive comments. The reviewer states that “The studies are thorough and of high quality and the authors provide convincing evidence to support their conclusions. I have only few minor comments that should be addressed prior to publication”.

1. The authors relay on inducing variable store depletion to control the number of STIM1-Orai1 puncta. Puncta number and Ca²⁺ influx are analyzed after 10 min treatment with thapsigargin. However, activation of transcription factors is analyzed 40 min after thapsigargin treatment. The authors should assay Ca²⁺ influx and puncta formation 40 min after thapsigargin treatment to ensure that the difference in Ca²⁺ influx and puncta formation is retained when transcription factors activation is measured.

We thank the reviewer for this important control. We have measured puncta formation and Ca²⁺ signals over 40 minutes and the new data are included in Figure 2 (Figure 2E) and as the new Supplemental Figure 2.

2. The results in Figure 3e,g are quite impressive and important. Should resolution permit, the authors should attempt similar binomial analysis of STIM1 puncta in Figure 2. This may provide further support for the relationship between STIM1 puncta numbers and NFAT1 activation.

We have spent some time over the past months trying to address this specific point. Following overexpression of STIM1-GFP, we measured both the number of puncta (shown in Figures 2D and 2E) and the fluorescent intensity of each punctum (using confocal microscopy). We found that the intensity of individual puncta followed a binomial distribution. 30 nM thapsigargin evoked fewer puncta, but the intensity of each punctum was similar to that induced by higher thapsigargin concentrations. However, we have not included the data because the spatial resolution afforded by confocal microscopy is ~250 nm, twice the size of a single ER-PM junction and probably several-fold larger than the size of an individual punctum. Therefore, we cannot be sure that we are not measuring several puncta at the same time. To circumvent this, we have used the Micron facility in the Biochemistry department at Oxford, which provides access to super-resolution microscopy. We have carried out SIM, which provides a spatial resolution of ~100 nm. However, although we see STIM1 puncta, analysis of the size/intensity of each punctum is riddled with problems. Very slight mis-alignments between images in a z-stack distort an individual punctum substantially (from our experience, by ~50%). Furthermore, individual puncta have complex geometry and cannot be represented by simple elliptical or spherical shapes. Currently available software (Image J) cannot

adequately analyze such varying shapes. We believe that it is for these reasons that studies that have shown puncta obtained from super-resolution microscopy have consistently failed to quantify individual puncta characteristics. We are currently collaborating with Dr Pradeep Kumar at Oxford Nanoimaging, an Oxford-based company that is developing a bench top super resolution microscope system. We hope to come up with a way to analyse puncta but this is a complex problem and we believe is outside the scope of the present study.

3. A control experiment should test the effect of loading the cells with BAPTA on STIM1-Orai1 clustering and the number of STIM1 puncta

Thank you for this comment. We have already reported (Singaravelu et al., JBC 286, 12189-12201) that loading cells with BAPTA had no effect on STIM1 redistribution and puncta formation after store depletion. We cite this in the paper.

Please provide statistics throughout in the Figures or legend (Figures 1g, i, 3b, e, g, 4b, 5b, h, 6c, e, j, 7d, e).

We have added statistics to the Figures and to the legends, as recommended.

Introduction ref 16: Some caution should be indicated in relation to claiming CRAC channels involvement in pancreatitis in view of the study reporting severe bacteremia and death by inhibition of acinar Orai1 (see PMID: 28273482).

Thank you for this important comment. To avoid a discussion of this issue in acinar cells, we have decided to omit the section from the Introduction.

Line 143: add of after each.

Thank you-corrected.

Line 291: no arrow is included in 3e.

Thank you-corrected.

Reviewer 4.

We thank the reviewer for his/her comments. The reviewer raised several issues.

1. Are there cases where the cell is depolarised with depleted stores (these are non-excitable cells)?

Yes, many cases have been described. In the same cell type we have used, the RBL cell line, it was reported several years ago that 100 nM thapsigargin, which causes store depletion and numerous puncta to form (Figures 1D-1G), also induces a strong depolarisation of the membrane potential (Evans et al. BBA 1718, 32-43).

There are numerous other examples of non-excitable cell with depolarised membrane potentials following physiological stimulation. This is particularly prominent with Gq-coupled receptors, which activate non-selective cation channels

of the TRP channel family and thereby depolarise the membrane potential rapidly. An elegant study by Penner, Kinet and colleagues demonstrated that agonist activated CRAC channels and the subsequent rise in cytosolic Ca^{2+} stimulated high conductance Na^{+} -permeable TRPM4 channels (Science 306, 1374-1377), leading to strong depolarization. Another nice example comes from studies in pancreatic acinar cells by Thorn and Petersen (JGP, 100, 11-25), who found that low concentrations of agonist opened a Ca^{2+} -dependent 25 pS non-selective cation channels in the basolateral membrane, leading to depolarization.

We have added an extra paragraph to the end of the discussion describing how physiological stimulation can cause membrane depolarisation as well as CRAC channel activation and thereby regulate Ca^{2+} flux through the store-operated channels.

2. Also they are comparing NFAT phosphorylation vs. c-fos transcription, two different processes with different timelines occurring different compartments in cells. It would be better if they studied the c-fos pathway using an upstream readout like activation of CREB (Sheng 1990) or STAT5 that also involves Ca dependent phosphorylation in the cytosol.

Thank you for this comment. The referee makes an important point and we have carried out several new experiments that address this concern directly. The new data are included as the new Figures 4D and 4E, the entire new Figure 5, the new Figure 6F and the new Supplemental Figure 5.

Specifically, we have carried out the following experiments.

- 1) We have compared expression of c-fos with that of an NFAT-dependent gene (interleukin 5; Kar, Mirams, Christian and Parekh Molecular Cell 64, 746-759) using quantitative real time PCR. High concentrations of thapsigargin increased transcription of both genes. However, stimulation with 100 nM thapsigargin in high K^{+} -containing external solution failed to increase c-fos expression (new Figure 4E) but caused a significant increase in interleukin 5 (new Figure 4D). These new data, using the same technique of qPCR, have been added to the manuscript.
- 2) As suggested by reviewer 4, we have studied an upstream readout of the c-fos pathway, namely Ca^{2+} -dependent activation of the transcription factor STAT5. Our previous work showed that local Ca^{2+} entry through CRAC channels activated the non-receptor tyrosine kinase Syk, tethered close to the CRAC channel (Samanta et al., Cell Reports 12, 203-216), which then phosphorylated STAT5 (Ng et al, JBC 284, 24767-24772). Once phosphorylated, STAT5 dimerises and migrates to the nucleus where it activates c-fos transcription (Darnell, Science 277, 1630-1635). We expressed STAT5-GFP and monitored its translocation from the cytosol to the nucleus following CRAC channel activation (Supplemental Figure 5). The new data demonstrate robust STAT5 migration to the nucleus following stimulation with high doses of thapsigargin (2 μM or 100 nM). However, stimulation with either 30 nM thapsigargin or 100 nM thapsigargin in high K^{+} -containing external solution failed to induce STAT5 movement compared with resting cells. This is in excellent agreement with the c-fos transcription data, where neither 30 nM thapsigargin nor 100 nM thapsigargin in high K^{+} -containing external solution increases c-fos expression (Figure 4E). By contrast, we find strong nuclear accumulation of NFAT-GFP in response to 100 nM thapsigargin

in high K⁺-containing external solution and modest but significant accumulation to 30 nM thapsigargin. Therefore, using fluorescence microscopy with GFP-tagged NFAT1 or GFP-tagged STAT5, we demonstrate that STAT5 is not activated under conditions that lead to strong stimulation of NFAT1. The STAT5 translocation data have been included as the new Supplemental Figure 5.

- 3) We have co-expressed RFP under an NFAT promoter together with a nuclear localized eYFP construct under a c-fos promoter. We have used FACS to quantify expression of RFP and eYFP in the same cells to the same stimulus. These data are included as the entire new Figure 5. The data show that neither 30 nM thapsigargin in normal K⁺-containing external solution nor 100 nM thapsigargin in high K⁺-containing external solution increase eYFP expression (indicating low c-fos activity) whereas RFP expression is high (indicating enhanced NFAT activity) in the same cells. We believe that this set of experiments, carried out in the same cells demonstrate unequivocally that the different transcription factors require different numbers of STIM1-Orai1 puncta as well as different extents of Ca²⁺ flux through the same number of puncta. The FACS studies provide data from >15,000 cells per condition. We have applied our new binomial spread function to these data and find they fit the empirical results very well. This new analysis is also included in Figure 5.

3. it is known that membrane depolarization plays a role in activating c-fos transcription (Sheng 1990), so that could be an additional confounding factor in their high K experiments.

We respectfully disagree with the reviewer here. The paper by Sheng, which the reviewer refers to, shows that high K⁺-containing external solution depolarises the membrane potential and activates c-fos transcription but this is accomplished through opening of voltage-gated Ca²⁺ channels. On page 481 of the Sheng paper (Neuron 4, 477-485), for example, it is stated:

“The influx of Ca²⁺ through voltage-dependent Ca²⁺ channels, leading to formation of the active Ca²⁺-calmodulin complex, appears to be the initial mode of activation of nicotine and other depolarizing stimuli such as elevated K⁺.”

The paper also points out that activation of IEGs (immediate early genes) by high K⁺ requires Ca²⁺ influx.

Sheng et al published a second paper in Neuron in 1990 demonstrating that high K⁺ activated c-fos transcription through membrane depolarization and subsequent opening of voltage-gated Ca²⁺ channels. The Ca²⁺ influx then stimulated CREB (Neuron 4, 571-582).

The elegant studies by Sheng, Greenberg and colleagues were carried out on excitable cells, which express voltage-gated calcium channels. High K⁺, which depolarises the membrane potential, opens these channels and the subsequent Ca²⁺ rise activates CREB followed by c-fos transcription. High K⁺ therefore activates c-fos through stimulation of voltage-dependent Ca²⁺ entry in excitable cells. Our work involves non-excitable RBL cells, which do not express voltage-gated calcium channels. This can be seen in patch clamp recordings (e.g. Figure 1B), which shows the complete absence of voltage-gated Ca²⁺ channels.

4. They could consider using patch clamping to hold the membrane potential instead of high K and use mCherry/RFP labeled NFAT in figure 3 experiments so they can do simultaneous Ca imaging with NFAT translocation.

We have already published simultaneous measurements of cytosolic Ca²⁺ (with fura 2) and NFAT-GFP translocation to the nucleus, following stimulation with either thapsigargin (Kar and Parekh Channels 7, 374-378) or after physiological stimulation with a low concentration of a Gq-coupled receptor agonist (see Figure 4 in Kar and Parekh, Molecular Cell 58 232-243). We have conducted patch clamp experiments in cells expressing NFAT1-GFP or NFAT1-cherry but we have not been able to maintain high quality seals with good series resistance for the 40 minutes required for full NFAT nuclear translocation.

5. Also, all experiments were performed using a RBL cell line that has known drawbacks in modeling human mast cells/basophils (Passante 2009), so they can consider including data from a different type of cell to make sure the results are generalizable.

We use RBL cells because they have a well-characterised and robust CRAC current and we have characterised CRAC channel-driven gene expression in this system in considerable detail over several years. They are therefore an ideal system for our purposes. It is not our aim in this particular study to focus on human mast cells although mast cell biology remains one of our primary interests and we study bone marrow-derived rodent mast cells in the laboratory (Lin, Kramer and Parekh, Molecular Cell 70, 228-241) and have access to human nasal tissue, from which we isolate nasal mast cells (Di Capite, Nelson, Bates and Parekh, Journal of Allergy and Clinical Immunology 124 1014-1021). We have opted, at this stage, to focus on depth on one system rather than breadth. However, we take the reviewer's comment on board and we are comparing the ability of 30 nM thapsigargin to activate NFAT and c-fos pathways with 100 nM thapsigargin in high K⁺-containing external solution, using primary mast cells. This is ongoing work and will be part of a more focussed study on mast cell biology. Preliminary data show that neither stimulus activates c-fos whereas both increase NFAT-dependent gene expression, with 100 nM thapsigargin in high K⁺-containing external solution being a stronger stimulus than 30 nM thapsigargin.

Minor comments:

· *There are not many references to the figure numbers throughout the text, more would be helpful for the reader to follow*

We have attempted to remedy this.

· *Line 176, please write concentration of high K solution*

This has been added.

*Figure legends refer to * representing p-values, but many bar graphs are missing * symbols*

Apologies for this; we have now rectified this.

- *Scale bars are needed for cell images*

These have been added.

- *Figure 2 does not have panels E and F as referred in the text (remove these in lines 245-249.)*

Apologies for this; this has been corrected.

- *Figure 2B, label for “rest” is duplicated*

Corrected.

- *Figure 3 E, G Thap 2 μM vs. 2000 nM; x-axis tick increment consistency*

Thank you for spotting this. It has been corrected (3G).

- *Figure 5-7: Y-axis label for NFAT is inconsistent, some are “Nuclear/Cytosol” and some are “N/C”*

Thank you for raising this. We have used N/C throughout.

- *Figure 6A cartoon is hard to understand without explanation*

Cartoon has been removed.

- *Line 267: missing period after first word*

Corrected.

- *Line 444: please include Hill-type equation*

Added to legend to Figure 8.

Reviewers' Comments:

Reviewer #2:

Remarks to the Author:

Authors properly responded to most of major questions raised by reviewers. A minor point is that there is still no proper statistics in figures 2 and 8.

Reviewer #3:

Remarks to the Author:

The authors addressed all my concerns and I have no further concerns.

Reviewer #4:

Remarks to the Author:

1. "It is also vital to directly measure CRAC currents in the low TG and high TG conditions to precisely determine how much Ca^{2+} is coming into the cell. Measuring cytosolic Ca^{2+} concentration does not equate current amplitude, which is what the authors are implying. This point was also raised by reviewer 2."

2. On page 6, the heading of the section says "Dependence of NFAT1 nuclear migration on STIM1 puncta number". Figure 3 shows the dependence on TG concentrations (not puncta number). As such, the data shown in the relevant figure does not show anything regarding STIM1.

3. Similarly, the section heading on page 8 says "Dependence of c-fos expression on the number of STIM1 puncta" Figure 4 that describes results in this section has no information on STIM1 puncta. In both cases, the dependence on STIM1 puncta is overstated.

We thank the reviewer for raising the additional points.

Figure 3. How do the images as depicted provide sufficient clarity and precision to determine varying degrees of nuclear/cytosol NFAT ratios? For example, the two images in the rightmost panels in A (30 nM TG and 100 nM TG/high K) supposedly show significantly different levels of N/C NFAT ratios. But from the fuzzy images shown, this quantification seems pretty difficult to draw. The quality of these wide-field images is extremely fuzzy to draw quantitative conclusions regarding differences in nuclear vs cytosolic NFAT between these conditions. Since this quantification is absolutely central to the point of the paper, this major technical weakness must be addressed. Confocal imaging would certainly address this, but the data shown in the paper is not obtained that way.

We thank the reviewer for this comment. We have repeated all NFAT experiments using confocal microscopy and analysed the new data. Confocal images have now been used throughout (Figures 3, 6 and 7) and the bar charts depict the new data.

1. "It is also vital to directly measure CRAC currents in the low TG and high TG conditions to precisely determine how much Ca²⁺ is coming into the cell. Measuring cytosolic Ca²⁺ concentration does not equate current amplitude, which is what the authors are implying. This point was also raised by reviewer 2."

Thank you for this comment. We have now carried out patch clamp recordings to measure CRAC current in low TG and high TG conditions. The new data have been included in Figure 1 (panels J and K).

2. On page 6, the heading of the section says "Dependence of NFAT1 nuclear migration on STIM1 puncta number". Figure 3 shows the dependence on TG concentrations (not puncta number). As such, the data shown in the relevant figure does not show anything regarding STIM1.

We have reworded the heading so it now refers to thapsigargin concentration.

3. Similarly, the section heading on page 8 says "Dependence of c-fos expression on the number of STIM1 puncta" Figure 4 that describes results in this section has no information on STIM1 puncta. In both cases, the dependence on STIM1 puncta is overstated.

We have reworded the heading so it now refers to thapsigargin concentration.

Reviewers' Comments:

Reviewer #4:

Remarks to the Author:

The authors have addressed previous concerns. I have no further comments.